# Leveraging Robust Optimization for LLM Alignment under Distribution Shifts

**Mingye Zhu**[1,2]    **Yi Liu**[2*]    **Zheren Fu**[1]
**Yongdong Zhang**[1]    **Zhendong Mao**[1]
[1]University of Science and Technology of China, Hefei, China
[2]State Key Laboratory of Communication Content Cognition, People's Daily Online, Beijing, China

## Abstract

Preference alignment methods are increasingly critical for steering large language models (LLMs) to generate outputs consistent with human values. While recent approaches often rely on synthetic data generated by LLMs for scalability and cost-efficiency reasons, this reliance can introduce distribution shifts that undermine the nuanced representation of human preferences needed for desirable outputs. In this paper, we propose a novel distribution-aware optimization framework that improves preference alignment despite such shifts. Our approach first leverages well-learned classifiers to assign a calibration value to each training sample, quantifying its alignment with the target human-preferred distribution. These values are then incorporated into a robust optimization objective that minimizes the worst-case loss over regions of the data space most relevant to human preferences. By explicitly focusing optimization on the target distribution, our approach mitigates the impact of distributional mismatch and improves the generation of responses that better reflect intended values.

## 1 Introduction

The rapid proliferation of large language models (LLMs) has made it increasingly important to ensure that model outputs align with human values. Techniques such as Reinforcement Learning from Human Feedback (RLHF) [33, 51, 40] and Direct Preference Optimization (DPO) [35, 45] have shown promise by leveraging high-quality, human-annotated data to guide model behavior [43, 2, 12]. A common approach to constructing alignment data involves either manual annotation by humans or the use of LLMs, with the former being resource-intensive and time-consuming, thus limiting scalability and broader applicability [4]. Consequently, recent research has increasingly focused on leveraging synthetic

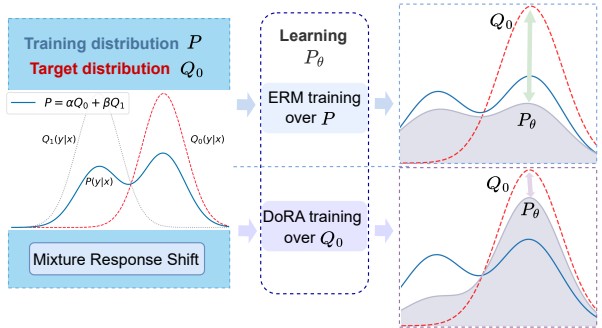

Figure 1: Comparison of ERM and DoRA Training. The left section illustrates the training distribution $P$, which is a mixture of human-preferred (target) distribution $Q_0$ and LLM distributions $Q_1$, highlighting the mixture response shift. The right section contrasts the outcomes of the traditional method (ERM training) over $P$ with the proposed DoRA training over $Q_0$, demonstrating how DoRA better aligns with the target distribution.

---

* Corresponding author: Yi Liu

39th Conference on Neural Information Processing Systems (NeurIPS 2025).

data generated by advanced LLMs, which have shown strong capability to simulate human preferences [22, 7, 8].

In this work, we concentrate on understanding and addressing one specific form of distribution shift arising from synthetic data within current alignment paradigms. As illustrated in Figure 1, traditional empirical risk minimization (ERM) assumes that the training distribution $P$ matches the target distribution $Q_0$, thus leading to a policy $P_\theta$ that aligns well with $Q_0$. However, in practice, $P$ does not perfectly mimic $Q_0$ due to two key limitations: (1) several studies suggest that training data derived from LLMs may exhibit inherent misalignment with human values [44, 14, 38, 42, 17], and (2) reward models are prone to biases [46, 47], potentially leading to suboptimal labelling. Consequently, the resulting training distribution $P$ may diverge significantly from the ideal distribution $Q_0$. In such settings, naïvely employing ERM training can yield a learned policy $P_\theta$ that is biased towards artifacts in the training data rather than genuinely aligning with human intent.

Mitigating distribution shifts in alignment data is a persistent yet understudied challenge. Existing robustness-focused approaches primarily address pairwise noisy labels and reward uncertainty [45, 6, 5], but they are tightly coupled with the Bradley-Terry (BT) model, limiting their versatility for broader alignment objectives (e.g., listwise optimization [48, 39, 30, 50]). To bridge this gap, we argue for more general robust optimization frameworks that not only mitigate distribution shifts from synthetic data but also seamlessly adapt to the growing complexity of alignment objectives.

In this paper, we propose **D**istribution-aware **o**ptimization for **R**obust **A**lignment (DoRA) to improve the robustness of alignment algorithms where training data comprises a mixture of heterogeneous sub-distributions—such as those arising from different synthetic sources or online updates [25, 29, 9, 18]. Rather than tailoring solutions to specific alignment formulations, our approach functions as a modular plug-in that enhances baseline robustness across diverse, and increasingly complex, alignment objectives encountered in real-world deployment. At the core of DoRA lies a simple yet effective strategy: it first employs well-trained distribution classifiers to assign a calibration score to each training sample, estimating its alignment with the target human-preferred distribution. These scores are then integrated into a distribution-aware optimization objective that minimizes the worst-case loss over data regions most representative of human preferences. This strategy ensures that the model remains resilient to distribution shifts between the training data and the target distribution, preventing it from disproportionately favoring biased synthetic patterns while still benefiting from their scalability.

> **📈 Contributions**
>
> We introduce DoRA, a distribution-aware optimization framework that robustly aligns LLM outputs with human preferences under distribution shifts. ***Technically***: We estimate the alignment of each sample with the target human-preferred distribution as a calibration term, which is then incorporated into a distribution-aware optimization objective that minimizes the worst-case loss over regions most relevant to human preferences. ***Theoretically***: We characterize the robust calibration alignment objective as a KL-divergence-based distributionally robust optimization problem, augmented with a reweighting mechanism. ***Emperically***: We demonstrate the effectiveness of our framework through extensive experiments, with consistent improvements in alignment metrics compared to state-of-the-art baselines.

## 2 Problem Formulation

**Notations**. Let $\mathbf{z} = \{x, y_0, \ldots, y_{n-1}\} \in Z$ represent a datum in a preference dataset $Z$, where each instance composes of one query $x$ and $n$ corresponding responses $y_0, y_1, y_2, \ldots, y_{n-1}$ . We assume that these observed data are drawn from a training distribution $P$, and the target distribution of interest, is denoted as $Q_0$. We then formalize the following definition.

**Definition 2.1** (Mixture Response Shift). A *Mixture Response Shift* occurs when, for any input query $x$, the conditional distribution of responses is a mixture of different distributions that partially overlaps with the target distribution $Q_0$ (with fraction $\alpha$), i.e.:

$$P(y|x) = \alpha\, Q_0(y|x) + \sum_{i=1}^{n-1} \beta_i\, Q_i(y|x), \tag{1}$$

where $\alpha, \beta_1, \ldots, \beta_{n-1} \geq 0$ and $\alpha + \beta_1 + \cdots + \beta_{n-1} = 1$.

***Remark*** 1. Definition 2.1 formalizes scenarios in which, for each given query $x$, a fraction $\alpha$ of the responses are drawn from the target distribution $Q_0$ (e.g., human preferred responses), while the remaining responses are sampled from other distributions $Q_1, \cdots . Q_{n-1}$(e.g., synthetic LLM generations). This setting reflects the practical reality that alignment training data often consists of a mixture of responses with diversified quality.

The core intuition is that training solely on data from the target distribution—as done in naïve supervised fine-tuning (SFT)—may limit generalization and fail to account for real-world variability. Robust alignment benefits from exposure to both diverse and suboptimal responses, which help models distinguish desirable behavior from undesirable patterns. When direct access to human-preferred data is limited, high-quality generations from strong models (e.g., GPT-4) can serve as a reasonable proxy of the golden responses from the target distribution.

**Distributionally robust optimization**: Distributionally Robust Optimization (DRO) [3, 11, 26] is a well-established framework that focuses on minimizing the worst-case expected loss over the perturbed or adversarial distribution $Q$ within the uncertainty set of $P$. Denote the instance-level loss as $\ell(\theta, \mathbf{z})$, then DRO is formulated as:

$$\min_{\theta \in \Theta} \sup_{Q \in \mathbb{P}} \mathbb{E}_Q[\ell(\theta, \mathbf{z})], \quad \mathbb{P} = \{Q \in \mathbb{D} : D(Q\|P) \leq \rho\}, \tag{2}$$

with $\mathbb{P}$ being the uncertainty set and $\mathbb{D}$ a set of all possible distributions. $D(Q\|P)$ is a distance metric, and $\rho$ is a parameter controlling the size of the ambiguity set. This objective aims to find $\theta$ that minimizes the expected loss over the worst-case distribution rather than minimizes the average performance over $P$. Specifically, we model the $\mathbb{P}$ using the Kullback-Leibler (KL) divergence, as it provides a tractable method for solving DRO problems [19]. With the change-of-measure technique [20], the *inner* maximization problem is then formally written as:

$$\max \mathbb{E}_P[h(\mathbf{z})\ell(\theta, \mathbf{z})], \quad \text{s.t.} \quad \mathbb{E}_P[h(\mathbf{z}) \log h(\mathbf{z})] \leq \rho, \tag{3}$$

where $h(\mathbf{z}) := \frac{dQ}{dP}(\mathbf{z})$ is the density ratio (Radon-Nikodym derivative) between $Q$ and $P$. Equation 3 can be viewed as a variational form of the problem in Equation 2, as it replaces the intractable optimization over distributions $Q$ with an optimization over the density ratio $h(\mathbf{z})$.

**Challenges:** While DRO methods are designed to handle sub-population shifts, they often suffer from over-pessimism [20, 49, 28], which manifests as poor generalization and overly conservative, low-confidence predictions, as excessive focus on worst-case scenarios can degrade overall performance.

## 3 Distribution-aware Optimization for Robust Alignment

To achieve more principled alignment and mitigate over-pessimism, we propose to extend the uncertainty set in the standard DRO framework to capture the mixture-induced shifts central to our setting. We then propose a calibration mechanism, derived from probabilistic classifiers, to infer the source distribution of each data point. Theoretically, we show that the proposed DoRA objective essentially optimizes a KL-based DRO objective, augmented with a distribution-aware reweighting mechanism. The overall framework is illustrated in Figure 2.

### 3.1 Sample-Level Calibration with Probabilistic Classifiers

In the KL-DRO framework, the worst-case target distribution can be written as $Q(\mathbf{z}) \propto P(\mathbf{z})e^{\eta\ell(\theta,\mathbf{z})}$ (See Appendix B.1 for derivation). This formulation implies that instances with higher loss values $\ell(\theta, \mathbf{z})$ are exponentially upweighted. While this mechanism enhances robustness by prioritizing challenging instances, it may also introduce over-pessimism when a small number of outliers (e.g., noisy or mislabeled points) have very large losses, thereby dominating the optimization.

To mitigate this issue, we introduce a simple calibration mechanism at the sample level. Rather than treating all samples equally when computing the robust loss, we scale the loss of each instance $\mathbf{z}$ by a data-dependent factor $\tilde{h}(\mathbf{z})$, which reflects its estimated relevance to the target distribution.

To formalize $\tilde{h}(\mathbf{z})$, we first define the family of $\alpha$-covered distributions based on Definition 2.1:

$$\mathbb{P}_\alpha = \left\{ Q_0 : P(y|x) = \alpha\, Q_0(y|x) + \sum_{i=1}^{n-1} \beta_i\, Q_i(y|x) \right\}. \tag{4}$$

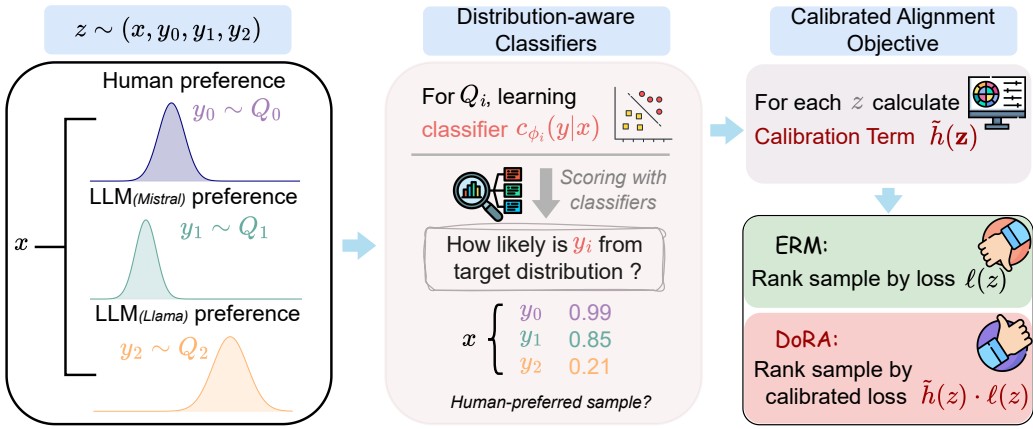

Figure 2: **The DoRA pipeline**. For each datum $\mathbf{z}$, where responses are drawn from a mixture of distributions, DoRA uses trained classifiers to estimate the alignment of each $y$ with the target distribution. These scores are then aggregated into a calibration term $\tilde{h}(\mathbf{z})$ for each sample, which reweights the original loss $\ell(\mathbf{z})$ during optimization to enable more principled robustness control.

Here, $P(y|x)$ denotes the observed distribution, and $Q_0(y|x)$ represents the target distribution. In classical DRO, the uncertainty set typically includes distributions that are "close" to a nominal empirical distribution $P$, often under some divergence measure. Here, we refine this setup by incorporating prior structural knowledge under response mixture shift.

**Learning distribution-aware classifiers**. To construct the calibration factor $\tilde{h}(\mathbf{z})$, we assume the existence of a "golden" (or reference) distribution, denoted $P_{\text{golden}}(y|x)$, provided by an oracle or approximated using trusted human-labeled data. Following Grover et al. [16], we train a probabilistic classifier $c_{\phi_i}$ for each sub-distribution $Q_i$ to estimate the probability that a response $y_i$ originates from the golden distribution. Under the assumption that $c_{\phi_i}$ is Bayes optimal, the *importance weight* for a response drawn from $Q_i$ is derived as:

$$w_{\phi_i}(y|x) = \frac{P_{\text{golden}}(y|x)}{Q_i(y|x)} = \gamma_i \frac{c_{\phi_i}(y|x)}{1 - c_{\phi_i}(y|x)}, \tag{5}$$

where $i \in \{1, \dots, n-1\}$. $\gamma_i$ is the *imbalance ratio* between a synthetic distribution and the golden distribution $\frac{q(y=0)}{q(y=1)}$. In practice, to estimate $P_{\text{golden}}$, we leverage human-preferred responses from the target distribution $Q_0$.

**Constructing the calibration factor.** We now construct an instance-dependent calibration factor $\tilde{h}(\mathbf{z})$ based on the mixture structure and the learned classifiers.

**Proposition 3.1.** *Let $P(y|x) = \alpha Q_0(y|x) + \sum_{i=1}^{n-1} \beta_i Q_i(y|x)$, with $\alpha + \beta_1 + \cdots + \beta_{n-1} = 1$ and $\alpha \in (0,1)$. Under the mixture response shift, we define $\tilde{h}(\mathbf{z})$ as an empirical estimate of the degree to which a given sample aligns with human preferences:*

$$\tilde{h}(\mathbf{z}) = \frac{1}{n}\Big(\frac{P_{golden}(y|x)}{\alpha\, Q_0(y|x)} + \cdots + \frac{P_{golden}(y|x)}{\beta_{n-1}\, Q_{n-1}(y|x)}\Big)$$
$$= \frac{1}{n}\Big(\frac{1}{\alpha}w_{\phi_0}(y|x) + \cdots + \frac{1}{\beta_{n-1}}w_{\phi_{n-1}}(y|x)\Big) = \frac{1}{n}\Big(\frac{\gamma_0 c_{\phi_0}(y|x)}{\alpha(1 - c_{\phi_0})} + \cdots + \frac{\gamma_{n-1} c_{\phi_{n-1}}(y|x)}{\beta_{n-1}(1 - c_{\phi_{n-1}})}\Big), \tag{6}$$

*where $w_{\phi_i}$ is defined earlier in Equation 5.*

***Remark*** 2. Proposition 3.1 introduces a *calibration factor* that can naturally incorporate mixture information into the robust objective. While the density ratio $h(\mathbf{z})$ quantifies how much a single sample is emphasized or downweighted in the adversarial distribution $Q$ compared to the nominal distribution $P$, the calibration term $\tilde{h}(\mathbf{z})$ serves as a proxy for human preference by estimating how likely a sample is to originate from the trusted component $Q_0$. Intuitively, $\tilde{h}(\mathbf{z})$ amplifies the contribution of responses that are more aligned with the golden distribution, and suppresses less preferred ones accordingly.

**Practical Considerations**. The importance weight in Equation 5 was originally introduced for binary classification tasks [41, 15]. When the learned classifier $c_{\phi_i}$ outputs probabilities that approach 1, the corresponding importance weights can become unbounded, leading to unstable and often problematic optimization. To rectify this, we insert a stabilizing term $\frac{1}{n}$ into each denominator, thereby bounding $\tilde{h}(\mathbf{z})$ such that $\tilde{h} : (\mathcal{X}, \mathcal{Y}) \to (0, n)$. Empirically, this simple modification prevents extreme weights when $c_{\phi_i}$ approaches 1, ensuring that no single sub-distribution excessively dominates training.

## 3.2 Deriving the DoRA objective

In this section, we derive the final alignment objective of DoRA. Specifically, we modify the pre-defined KL-DRO framework in Equation 3 by modulating each sample's contribution by a precomputed calibration factor $\tilde{h}(z)$, which reflects its estimated alignment with the target distribution. The robust objective is defined as:

$$\max \mathbb{E}_P[h(\mathbf{z})\tilde{h}(\mathbf{z})\ell(\theta, \mathbf{z})], \quad \text{s.t.} \quad \mathbb{E}_P[h(\mathbf{z})\log h(\mathbf{z})] \le \rho, \tag{7}$$

Since $\tilde{h}(\mathbf{z})$ is a fixed scalar per instance, it does not affect the optimization variables in the Lagrangian. Consequently, the worst-case risk from Equation 7 admits a closed-form dual solution:

**Proposition 3.2** (Worst-case risk under mixture response shift). *Let the loss be modulated by instance-level weights $\tilde{h}(\mathbf{z})$, which are fixed and known. Then the worst-case risk under a KL constraint $\rho$ is given by:*

$$\mathcal{R}(\theta) = \inf_{\lambda > 0} \left\{ \lambda \log \mathbb{E}_P \left[ \exp \frac{1}{\lambda} \left( \tilde{h}(\mathbf{z}) \cdot \ell(\theta, \mathbf{z}) \right) \right] + \lambda \rho \right\}. \tag{8}$$

*In turn, minimizing $\mathcal{R}(\theta)$ over $\theta$ reduces to*

$$\min_{\theta \in \Theta} \lambda \log \mathbb{E}_P \left[ \exp \frac{1}{\lambda} \left( \underbrace{\tilde{h}(\mathbf{z})}_{\text{Calibration Term}} \ell(\theta, \mathbf{z}) \right) \right], \tag{9}$$

*where $\mathbf{z}$ is an instance of $(x, y_0, \cdots, y_{n-1})$.*

***Remark*** 3. See Appendix B.2 for the detailed Lagrangian derivation. The calibration term $\tilde{h}(\mathbf{z})$ acts as a soft indicator of how closely each data sample's responses (for a given query $x$) align with the "ideal" or golden distribution, thereby determining its relative importance during training. Consequently, if a sample has a large loss $\ell(\theta, \mathbf{z})$ but a small $\tilde{h}(\mathbf{z})$, DoRA deems it less valuable to learn from and downweights it accordingly. Please see Algorithm 1 for more details.

---

**Algorithm 1** DoRA Optimization Algorithm

---

**Require:** Pretrained model parameters $\theta$, robustness parameter $\lambda$, dataset $\mathcal{D} = \{\mathbf{z}_i = (x_i, y_{i,0}, \ldots, y_{i,n-1})\}_{i=1}^N$

**Ensure:** Optimized model parameters $\theta$

    ▷ **Phase 1: Classifier Learning**

1: **for** $j = 0 \ldots n - 1$ **do**
2:     Assign binary labels:

$$l_i^{(j)} = \begin{cases} 1 & \text{if } y_{i,j} \text{ is target,} \\ 0 & \text{otherwise} \end{cases}$$

3:     Train classifier $c_{\phi_j}$ using $\{(x_i, y_{i,j}, l_i^{(j)})\}_{i=1}^N$
4: **end for**

    ▷ **Phase 2: DoRA Training**

1: Precompute $\tilde{h}(\mathbf{z}_i) \forall \mathbf{z}_i \in \mathcal{D}$ via Eq. 6
2: **while** not converged **do**
3:     Sample batch $\mathcal{B} = \{\mathbf{z}_i\}_{i=1}^b \subset \mathcal{D}$
4:     Compute robust loss over $\mathcal{B}$ according to Equation 9:

$$\mathcal{L} = \lambda \log \left( \frac{1}{|\mathcal{B}|} \sum_{i=1}^b \exp \left( \frac{\tilde{h}(\mathbf{z}_i)\ell(\theta, \mathbf{z}_i)}{\lambda} \right) \right)$$

5:     Update $\theta \leftarrow \theta - \eta \nabla_\theta \mathcal{L}_{\text{DoRA}}$
6: **end while**

---

## 4 Experiments

**Models and Datasets**. We validate the proposed method with two base models: Mistral-7B-v0.1, Llama-3.1-8B, on three widely used datasets in alignment literature: HH-RLHF, Summarization and

the UltraFeedback datasets. Specifically, we consider the following settings: 1.**pairwise** preference setting where we leverage the original pairwise data; 2. **listwise** preference setting, where we augment the original pairwise data with 2 additional synthetic responses from Mistral-7B-Instruct-v0.3, leading to 4 responses in total for each query. We primarily focus on *Setting 2* as we believe they represent a more practical scenario for the current alignment paradigm where models learn from multiple higher-quality samples from various sources.

**Baselines**. For pairwise comparisons, we employ four well-performed baselines: **DPO** [35], **R-DPO** [34], **EXO** [21] and **SimPO** [31]. For listwise contrasts, we use $\text{DPO}_{\text{PL}}$[2] [35], **RRHF** [48] and **LIRE** [50]. Please find the detailed objectives for these algorithms in Appendix C.5.

**Experimental Settings**. For each task, we first train classifiers as specified in Algorithm 1 with a simple BERT base model. The chosen response in the original dataset is labeled as 1 and the other as 0. The well-trained classifiers output probabilities that indicate the likelihood score that a response belongs to the target distribution. We also train an SFT model on the preferred responses to serve as the starting point before policy optimization. We set $\lambda = 1$ for all experiments. Detailed hyperparameter configurations and additional training settings are provided in Appendix C.

### 4.1 Experimental Results

**Performance on pairwise preference datasets.** For all four pairwise baselines, we use the original preference datasets without any response augmentation. As illustrated in Figure 3, DoRA consistently improves performance across all baselines. This suggests that even widely adopted "golden" datasets are affected by distribution shifts or contain noisy samples.

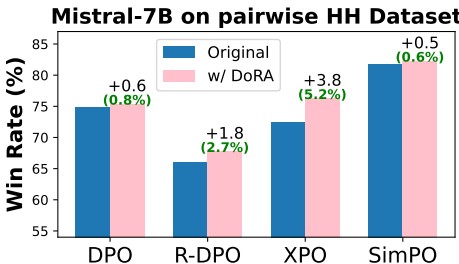 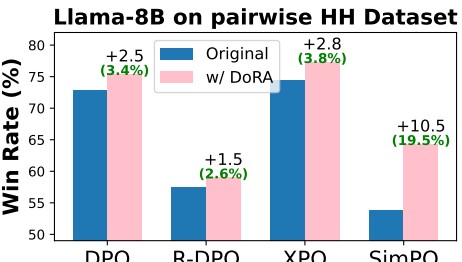

Figure 3: **DoRA boosts pairwise baseline performance**. When applied to unaugmented "golden" preference datasets, DoRA consistently enhances response quality across all baselines.

**Comparison with robust baselines.** Robust DPO [6] and Dr.DPO [45] are two strong baselines designed to address label-flip noise in pairwise preference data, particularly in BT preference models. Since both methods operate in the pairwise setting, we compare their performance against the DoRA-aligned DPO variant with HH dataset in Table 1. It is important to note that DoRA is a *method-agnostic* module that can be seamlessly integrated into a wide range of alignment objectives—not limited to the pairwise case. From the results, we observe that all robust baselines outperform vanilla DPO, confirming the value of robustness in preference learning. Notably, DoRA achieves the lowest lose rates among all methods, suggesting that its calibration mechanism provides more flexible and fine-grained control.

Table 1: **Comparison with pairwise robust baselines on HH dataset**. While all the approaches improve the win rate of vanilla DPO, DoRA achieves the lowest lose rates compared to the counterparts.

| Baselines | Mistral-7B | | Llama-8B | |
|---|---|---|---|---|
| | Win(↑) | Lose(↓) | Win(↑) | Lose(↓) |
| DPO [35] | 74.8 ($\pm$1.84) | 23.5 ($\pm$1.80) | 72.9 ($\pm$1.02) | 23.7 ($\pm$1.65) |
| Robust DPO [6] | **77.8** ($\pm$2.20) | 20.0 ($\pm$2.10) | 74.2 ($\pm$1.43) | 22.9 ($\pm$1.23) |
| Dr.DPO [45] | 75.2 ($\pm$1.64) | 22.4 ($\pm$1.63) | 73.0 ($\pm$1.41) | 24.3 ($\pm$0.71) |
| DPO w/ DoRA | 75.4 ($\pm$1.61) | **21.4** ($\pm$1.46) | **75.4** ($\pm$0.89) | **20.9** ($\pm$0.67) |

---

[2]While pairwise DPO is derived under the Bradley-Terry family of preference models in particular, listwise $\text{DPO}_{\text{PL}}$ is derived under more general Plackett-Luce preference model to handle multiple reward signals.

**Performance on listwise preference datasets.** Next, we experiment with the augmented datasets, which directly simulate a mixture response shift scenario which candidate responses are drawn from diverse underlying distributions. As shown in Table 2, incorporating DoRA leads to performance gains consistently across multiple baselines and tasks, with higher win rates and lower lose rates against the reference responses, underscoring its effectiveness in mitigating distributional shifts.

Table 2: **DoRA enhances listwise baselines on dialogue and summarization tasks**. *Win* indicates that GPT-4o assesses DoRA's response as superior compared to the golden responses from the datasets. **Bold** numbers suggest DoRA the winner. The results demonstrate that incorporating DoRA generally improves performance or at least keeps it on par with the baselines.

| Alignment | HH-RLHF | | | | Summarization | | | |
|---|---|---|---|---|---|---|---|---|
| | Mistral-7B | | Llama-8B | | Mistral-7B | | Llama-8B | |
| | Win(%)↑ | Lose(%)↓ | Win(%)↑ | Lose(%)↓ | Win(%)↑ | Lose(%)↓ | Win(%)↑ | Lose(%)↓ |
| DPO$_{PL}$ [35] | 75.0 | 22.5 | 81.0 | 18.0 | 53.3 | 46.3 | 54.5 | 42.8 |
| w/ **DoRA** | **78.0** | **21.0** | **82.5** | **16.0** | **55.3** | **43.5** | **59.0** | **30.5** |
| RRHF [48] | 76.5 | 19.5 | 43.8 | 56.0 | 70.0 | 29.5 | 70.8 | 28.8 |
| w/ **DoRA** | **79.8** | **19.0** | **44.5** | **55.0** | **72.0** | **28.0** | **74.0** | **25.0** |
| LIRE [50] | 72.8 | 26.8 | 82.0 | 17.5 | **82.5** | **17.5** | 82.5 | 17.0 |
| w/ **DoRA** | **84.0** | **16.0** | **84.5** | **14.5** | 81.0 | 19.0 | **83.8** | **16.0** |

To further evaluate the robustness of our method, we benchmark models trained on the augmented UltraFeedback dataset using AlpacaEval 2 [10] and Arena-Hard [23]. As reported in Table 3, DoRA generally improves the instruction-following capabilities for the baselines, particularly for LIRE. However, we observe a slight performance drop for DPO$_{PL}$, which may be attributed to the nature of the UltraFeedback dataset. Specifically, the chosen responses in UltraFeedback are drawn from a mix of distributions themselves, deviating from our core assumption that they originate from a single source. This distributional mismatch could reduce the effectiveness of the learned classifier, thus impacting overall performance.

Table 3: **AlpacaEval 2 and Arena-Hard results**. Experiments suggest that DoRA keeps or improves the instruction-following capabilities of baselines trained on the augmented UltraFeedback dataset.

| Dataset | AlpacaEval 2.0 | | | Arena-Hard |
|---|---|---|---|---|
| Metric | LC(%) | WR(%) | Length | WR(%) |
| DPO$_{PL}$ | **18.80** | **18.14** | 1972 | **12.3** |
| w/ **DoRA** | 18.27 | 17.73 | 1943 | 12.2 |
| RRHF | 10.52 | 13.88 | 1494 | 11.0 |
| w/ **DoRA** | **10.65** | **14.48** | 1446 | **11.2** |
| LIRE | 28.74 | 29.44 | 1815 | 20.0 |
| w/ **DoRA** | **31.28** | **32.02** | 1972 | **20.3** |

**DoRA enhances reward-confidence correlation for reliable generation.** Figure 4 plots normalized reward scores against classifier confidence probabilities for generated responses, where each point represents one model output. The dashed lines represent best-fit linear relationship between normalized reward scores and classifier probabilities, derived via ordinary least squares regression. The figure demonstrates that DoRA displays more consistent alignment between the classifier's confidence and the normalized rewards, indicated by a clearer positive slope in the regression line. This suggests that DoRA facilitates a more reliable prediction of outcomes, where higher rewards correspond more likely to higher classifier probabilities. Moreover, we see overall reward distribution of DoRA shifts rightward, especially for Llama3 model, indicating more high-quality responses.

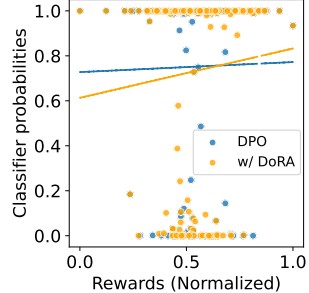
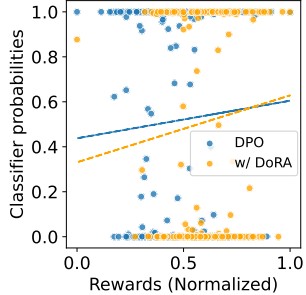

(a) Mistral-7B on HH dataset    (b) Llama-8B on HH dataset

Figure 4: **Reward-confidence correlation for generated responses**. DoRA exhibits stronger reward-confidence calibration than baselines, evidenced by steeper regression slopes ($\Delta\beta$=+0.174 for Mistral and $\Delta\beta = +0.131$ for Llama, larger slope means better correlation). This indicates DoRA's high-reward outputs more closely match the target distribution's characteristics, validated by elevated classifier probabilities.

## 4.2 Ablations

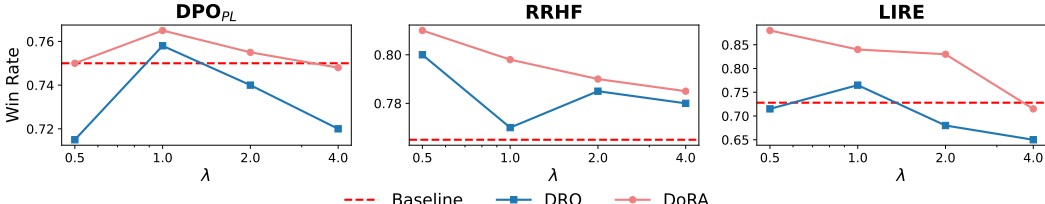

Figure 5: **Performance variation with different choices of $\lambda$ for vanilla DRO and DoRA**. We observe that as $\lambda$ increases from 0.5 to 4.0, the win rate generally decreases, albeit with some variations. Besides, vanilla DRO generally downperforms the proposed DoRA.

**Impact of $\lambda$ on robustness and performance.** The regularization parameter $\lambda$ in DoRA controls the balance between flexibility and robustness. Specifically, smaller $\lambda$ values induce sharper exponential weighting, emphasizing high-loss samples and increasing robustness. Larger $\lambda$ values reduce this effect, making the objective closer to standard ERM training. Figure 5 presents that $\lambda = 1$ yields strong performance on the HH task with Mistral-7B, while increasing $\lambda$ slightly decreases the win rate, underscoring the importance of tuning robustness. Figure 6 further illustrates that larger $\lambda$ yields policies whose behavior more closely resembles that of the baseline ERM-trained model.

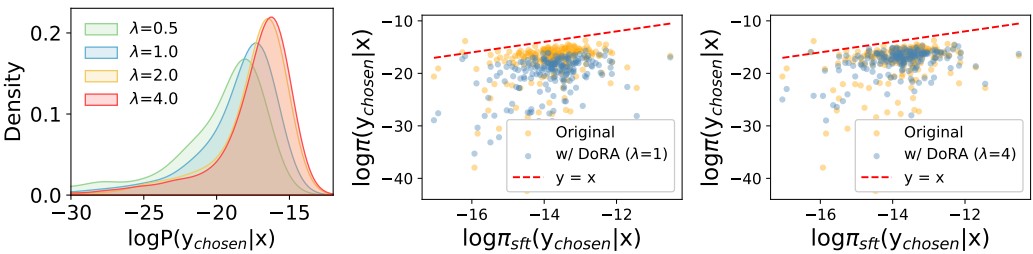

(a) DoRA w/ different $\gamma$ values     (b) Log prob. distribution w/ $\lambda = 1$   (c) Log prob. distribution w/ $\lambda = 4$

Figure 6: **Impact of different $\lambda$ values on policy distribution**. (a). Larger $\lambda$ increases model probabilities on the HH-chosen responses. (b/c). Scatter plots comparing log probabilities assigned to HH-preferred responses. The x-axis shows log probabilities under the initial SFT policy; the y-axis shows those under the original baseline and DoRA policies, respectively. As $\lambda$ increases, DoRA shifts closer to the original baseline (ERM training), assigning higher likelihoods to preferred responses.

**Effectiveness of the DRO framework and reweighting mechanism.** We investigate the isolated effects of the DRO formulation and the reweighting mechanism across different alignment objectives. As shown in Table 4, both approaches can yield performance improvements, but their effectiveness is highly dependent on the specific alignment loss and task. For instance, methods like DPO appear relatively robust to how the loss is handled,

Table 4: **Comparison with DRO and reweighting mechanism**. The results show that combining calibration-aware scoring and robust optimization generally brings out better performance.

| Strategy | Expression | Loss $\ell(\theta, \mathbf{z})$ | HH Data | Sum. Data |
|---|---|---|---|---|
| DRO | $\log \mathbb{E}_{\mathbf{z}} \exp \frac{\ell(\theta,\mathbf{z})}{\lambda}$ | $DPO_{PL}$ | 76.0 | 54.0 |
| | | RRHF | 77.0 | 52.8 |
| | | LIRE | 65.0 | 70.0 |
| Reweighting | $\tilde{h}(\mathbf{z})\ell(\theta, \mathbf{z})$ | $DPO_{PL}$ | 74.5 | 55.0 |
| | | RRHF | **80.5** | 48.3 |
| | | LIRE | 80.8 | 78.3 |
| DoRA | $\tilde{h}(\mathbf{z}) \log \mathbb{E}_{\mathbf{z}} \exp \frac{\ell(\theta,\mathbf{z})}{\lambda}$ | $DPO_{PL}$ | **78.0** | **55.3** |
| | | RRHF | 79.8 | **72.0** |
| | | LIRE | **84.0** | **81.0** |

showing only modest fluctuations across variants. In contrast, LIRE demonstrates a higher sensitivity to robustness techniques, with substantial performance gains or drops depending on the strategy used. In general, we conclude from the results that the combination of calibration-aware scoring and robust optimization is more effective than applying either component

alone, as their synergy leads to better distributional alignment and more reliable preference modeling under real-world distributional shifts.

## 4.3 Generalization and Robustness Evaluation

**Label noise**. In the preceding experiments, all models were trained on clean datasets without synthetic perturbations. To further assess the robustness of DoRA, we conduct corrupted-label experiments that serve as stress tests under adversarial distribution shifts. In particular, label noise is introduced by randomly flipping a proportion of labels in the original training data. Experimental results indicate that the incorporation of DoRA consistently enhances model robustness as the corruption rate increases, leading to stable and significant improvements across multiple benchmarks. This further demonstrates DoRA's effectiveness and reliability in real-world applications where annotation noise or distributional shifts are inevitable.

Table 5: Robustness evaluation under different corruption rates on HH and AlpacaEval 2 datasets. DoRA consistently improves robustness across methods.

| Dataset | HH-RLHF | | | AlpacaEval 2 | |
|---|---|---|---|---|---|
| Corruption rate | 20% | 40% | 60 % | 40 % | 40 % |
| Metric | WR(%) | | | LC(%) | WR(%) |
| DPO$_{PL}$ | 71.0 | 64.5 | 57.3 | 11.0 | 11.0 |
| w/ DoRA | 74.5 | 67.0 | 60.8 | 12.1 | 12.1 |
| RRHF | 63.5 | 42.3 | 26.3 | 7.7 | 4.8 |
| w/ DoRA | 65.5 | 44.5 | 30.8 | 8.5 | 5.5 |
| LIRE | 67.7 | 52.5 | 61.4 | 26.5 | 25.1 |
| w/ DoRA | 71.5 | 55.0 | 65.3 | 27.7 | 26.2 |

**Self-training**. While the primary focus of this work is on mixture response shifts induced by synthetic data in offline training, we consider online adaptation as a promising direction. To provide preliminary evidence, we conduct a small-scale experiment where we first train LLaMA-3.2-3B on the HH dataset (Iterate 1), and then continue training it on its own generated responses for two additional iterations. This setup naturally induces an evolving mixture shift, as the response distribution becomes increasingly synthetic. The results demonstrate that DoRA remains robust under such conditions, indicating that its instance-level calibration generalizes well to online or self-generated data scenarios where distributional drift arises organically.

Table 6: Performance improvement across training iterations. DoRA consistently enhances final performance under self-training scenario.

| Method | Iter 1 | Iter 2 | Iter 3 |
|---|---|---|---|
| DPO$_{PL}$ | 77.0 | 83.5 | 85.0 |
| w/ DoRA | 78.5 | 85.8 | 88.0 |
| RRHF | 45.8 | 47.8 | 50.5 |
| w/ DoRA | 47.5 | 49.3 | 52.5 |
| LIRE | 80.3 | 83.0 | 84.5 |
| w/ DoRA | 82.5 | 85.0 | 86.8 |

## 5 Related Works

**Preference Alignment for LLMs.** Since LLMs are pre-trained on vast internet data, they can generate outputs that are biased, harmful, or misaligned with human values. To address this, preference alignment techniques have emerged as key solutions. RLHF utilizes a reward model trained on human feedback to guide reinforcement learning, and DPO streamlines the process by directly optimizing the model to prefer desirable responses without an explicit reward model. Building on these approaches, recent research has proposed refinements to improve alignment efficiency and robustness. For example, Azar et al. [1] presents a generalized preference optimization framework, Ethayarajh et al. [13] introduces a novel loss function for enhanced robustness, and Meng et al. [31] explores simplified objectives to reduce computational overhead. These advancements reflect the ongoing effort to develop more scalable and effective alignment methods.

**Synthetic Data for Alignment.** Preference alignment typically relies heavily on human-annotated datasets, but the high cost and limited scalability of such data present a major bottleneck. To address this, recent research has explored leveraging synthetic data for alignment. For example, RLAIF [22] synthesizes preference data and uses PaLM 2 for feedback, while UltraFeedback [7] employs GPT-4 to annotate LLM-generated responses, creating scalable training datasets. Moreover, researchers have integrated synthetic data to expand candidate pools for preference learning [39, 48, 30, 50], demonstrating the potential of synthetic data. However, the use of synthetic data introduces new challenges. One key issue is the mismatch between the sampling distribution and the learning policy. To address this, RSO [29] employs rejection sampling to source preference data from the estimated

target optimal policy, thereby improving the accuracy of the maximum likelihood estimator. Another critical challenge lies in the distributional inconsistencies between synthetic and human-generated data during preference learning. This shift can hinder alignment performance, leading to biased model behaviors to true human preferences[27]. In this paper, we focus on tackling this latter challenge, aiming to enhance the robustness of preference learning in the presence of distribution shifts.

**Robustness in Alignment.** DRO [19] is a well-established framework that minimizes the worst-case training loss over a set of pre-defined groups, ensuring robustness to distributional shifts. In language modeling, Oren et al. [32] applies DRO to minimize losses over worst-case topic mixtures, while Sagawa et al. [37] enhances worst-group generalization in overparameterized regimes through increased regularization. For preference alignment, robust optimization techniques are explored to address challenges like reward uncertainty and noisy data. MaxMin-RLHF [5] learns a mixture of reward functions via expectation maximization to cater to diverse human preferences. ROPO [24], Robust DPO [6] and Dr.DPO [45] focus on noise tolerance in the pairwise BT paradigm. For instance, ROPO derives a robust loss by suppressing the gradients of samples with high uncertainty, and Dr.DPO optimizes against worst-case pairwise scenarios for DPO. Similarly, GRPO [36] builds upon reward-free DPO method by prioritizing groups with worse cumulative loss iteratively. In contrast to the prior work that focuses on label noise or is designed under the BT framework, our work aims to learn a method-agnostic approach that may seamlessly generalize to diverse alignment paradigms.

# 6 Discussion

**Conclusion.** In this paper, we propose a distribution-aware robust alignment framework that alleviates the influence of synthetic data bias and distribution shifts in LLM alignment. By leveraging a learned classifier to aggregate a calibration term to the DRO objective, DoRA effectively balances the scalability of synthetic data with the fidelity of human-aligned outputs.

**Limitation and future work.** Our framework assumes access to coarse-grained information about the source of the data (e.g., human- or model-generated) to guide classifier training. While this is often feasible in curated alignment pipelines, it may be less accessible in fully open-domain or legacy datasets. Moreover, DoRA presumes that the target distribution can be at least approximately estimated or measured; in scenarios where such reference distributions are unavailable or unreliable, its applicability may be limited. Identifying principled heuristics or proxy objectives for such cases represents an important direction for future work. Exploring ways to relax this requirement through unsupervised or weakly supervised signals presents an exciting direction for future research. Our use of DRO is particularly beneficial in scenarios with distribution shifts or variable alignment quality. While its conservativeness may offer limited gains in clean settings, it holds promise in reliability-critical domains such as factual generation or value-sensitive applications. Finally, although online adaptation is briefly mentioned in Section 4.3, extending DoRA to broader online learning scenarios remains a promising direction toward continual adaptation and scalability.

## Acknowledgments and Disclosure of Funding

This research is supported by Artificial Intelligence-National Science and Technology Major Project 2023ZD0121200 and the National Science Fund for Excellent Young Scholars under Grant 62222212. Besides, we are grateful to Jiashuo Liu and Junkang Wu for providing helpful discussions during the preparation of this paper.

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

# A   Broader Impacts

As LLMs grow more capable, they also pose escalating risks—including misinformation, deception, bias, and harmful content—with potentially severe societal consequences. To steer model outputs toward human values and intentions, developing robust techniques for ethical alignment has become imperative. Significant research efforts now target ethical AI frameworks spanning data curation, algorithmic design, and deployment safeguards. Our work aims to advance this critical frontier, enhancing the safety and robustness of LLMs for real-world applications.

# B   Mathematical Derivation

## B.1   The worst-case distribution in KL-DRO

In KL-DRO, we consider the following optimization problem:

$$\max_Q \mathbb{E}_Q[\ell(x)] \quad \text{subject to} \quad D_{KL}(Q\|P) \le \rho,$$

where $\ell(x)$ is the loss function. Let $\lambda \ge 0$ be a Lagrange multiplier for the KL constraint, we have the following Lagrangian relaxation:

$$\mathcal{L} = \mathbb{E}_Q[\ell(x)] - \lambda\left(D_{KL}(Q\|P) - \rho\right),$$

then substitute $D_{KL}(Q\|P) = \mathbb{E}_Q\left[\ln\frac{Q(x)}{P(x)}\right]$:

$$\mathcal{L} = \int Q(x)\ell(x)\,dx - \lambda\left(\int Q(x)\ln\frac{Q(x)}{P(x)}\,dx - \rho\right).$$

Vary $\mathcal{L}$ with respect to $Q(x)$. The functional derivative is:

$$\frac{\delta\mathcal{L}}{\delta Q(x)} = \ell(x) - \lambda\left(\ln\frac{Q(x)}{P(x)} + 1\right).$$

Set this derivative to zero:

$$\ell(x) - \lambda\left(\ln\frac{Q(x)}{P(x)} + 1\right) = 0.$$

We are left with

$$\frac{Q(x)}{P(x)} = e^{\frac{\ell(x)}{\lambda} - 1}.$$

The Lagrange multiplier $\lambda$ is implicitly tied to $\rho$. For simplicity, redefine $\eta \to \frac{1}{\lambda}$, leading to:

$$Q(x) \propto P(x)e^{\eta\ell(x)}.$$

## B.2   Dual Optimization

We proceed with the robust objective in Equation 7. First we introduce a Lagrange multiplier $\lambda \ge 0$ for the KL constraint and $\mu \in \mathbb{R}$ for the normalization constraint $\mathbb{E}_P[h] = 1$. The Lagrangian becomes:

$$\mathcal{L}(h, \lambda, \mu) = \mathbb{E}_P\left[h(\mathbf{z}) \cdot \tilde{h}(\mathbf{z}) \cdot \ell(\theta, \mathbf{z})\right] - \lambda\left(\mathbb{E}_P[h(\mathbf{z})\log h(\mathbf{z})] - \rho\right) - \mu\left(\mathbb{E}_P[h(\mathbf{z})] - 1\right)$$

$$= \mathbb{E}_P\left[h(\mathbf{z}) \cdot \left(\tilde{h}(\mathbf{z})\ell(\theta, \mathbf{z}) - \lambda\log h(\mathbf{z}) - \mu\right)\right] + \lambda\rho + \mu$$

To find the optimal $h^*$, we solve the variational problem by setting the functional derivative of $\mathcal{L}$ with respect to $h(\mathbf{z})$ to zero:

$$\frac{\delta\mathcal{L}}{\delta h(\mathbf{z})} = \tilde{h}(\mathbf{z}) \cdot \ell(\theta, \mathbf{z}) - \lambda(1 + \log h(\mathbf{z})) - \mu = 0$$

Solving for $h(\mathbf{z})$, we get:

$$h^*(\mathbf{z}) = \exp\left(\frac{1}{\lambda}\tilde{h}(\mathbf{z}) \cdot \ell(\theta, \mathbf{z}) - 1 - \frac{\mu}{\lambda}\right)$$

We denote the normalizing constant:

$$Z := \mathbb{E}_P\left[\exp\left(\frac{1}{\lambda}\tilde{h}(\mathbf{z}) \cdot \ell(\theta, \mathbf{z})\right)\right] \quad \Rightarrow \quad h^*(\mathbf{z}) = \frac{1}{Z}\exp\left(\frac{1}{\lambda}\tilde{h}(\mathbf{z}) \cdot \ell(\theta, \mathbf{z})\right)$$

Now substitute $h^*$ back into the original objective:

$$\mathbb{E}_P[h^*(\mathbf{z}) \cdot \tilde{h}(\mathbf{z}) \cdot \ell(\theta, \mathbf{z})] = \frac{1}{Z}\mathbb{E}_P\left[\tilde{h}(\mathbf{z}) \cdot \ell(\theta, \mathbf{z}) \cdot \exp\left(\frac{1}{\lambda}\tilde{h}(\mathbf{z}) \cdot \ell(\theta, \mathbf{z})\right)\right]$$

Now the simplified dual expression can be written as:

$$\inf_{\lambda > 0}\left\{\lambda\rho + \lambda\log\mathbb{E}_P\left[\exp\left(\frac{1}{\lambda}\tilde{h}(\mathbf{z}) \cdot \ell(\theta, \mathbf{z})\right)\right]\right\}$$

### B.3   Convergence Analysis

In this section, we analyze the convergence properties of the DoRA formulation. In particular, we show that under suitable conditions on the loss function (convexity and smoothness) and the density ratios, our robust objective converges to a global optimum via gradient-based methods.

**Robust Objective Formulation**. First we define

$$f(\theta) = \log\left(\mathbb{E}_P\left[\exp\left(\frac{l(\theta, \mathbf{z})}{\lambda}\right)\right]\right).$$

The function $f(\theta)$ is the well-known log-sum-exp (LSE) function, which is a smooth convex approximation of the maximum. Assume that for every $\mathbf{z}$, the loss function $l(\theta, \mathbf{z}) = \tilde{h}(\mathbf{z}) \cdot \ell(\theta, \mathbf{z})$ is convex in $\theta$ and has a Lipschitz continuous gradient with constant $L$:

$$\|\nabla l(\theta_1, \mathbf{z}) - \nabla l(\theta_2, \mathbf{z})\| \leq L\|\theta_1 - \theta_2\|, \quad \forall\theta_1, \theta_2.$$

Since the exponential function is convex and increasing, the mapping

$$\theta \mapsto \exp\left(\frac{l(\theta, \mathbf{z})}{\lambda}\right)$$

is convex for each $\mathbf{z}$. Taking the expectation over $P$, we obtain that

$$g(\theta) = \mathbb{E}_P\left[\exp\left(\frac{l(\theta, \mathbf{z})}{\lambda}\right)\right]$$

is convex in $\theta$. Moreover, since the logarithm is a monotonic transformation, $f(\theta) = \log g(\theta)$ is also convex. The gradient of $f(\theta)$ is given by

$$\nabla f(\theta) = \frac{1}{\lambda}\frac{\mathbb{E}_P\left[\exp\left(\frac{l(\theta,\mathbf{z})}{\lambda}\right)\nabla l(\theta, \mathbf{z})\right]}{\mathbb{E}_P\left[\exp\left(\frac{l(\theta,\mathbf{z})}{\lambda}\right)\right]}.$$

This expression can be interpreted as a weighted average of $\nabla l(\theta, \mathbf{z})$, where the weights

$$\tilde{p}_\theta(\mathbf{z}) = \frac{\exp\left(\frac{l(\theta,\mathbf{z})}{\lambda}\right)}{\mathbb{E}_P\left[\exp\left(\frac{l(\theta,\mathbf{z})}{\lambda}\right)\right]}$$

form a softmax distribution. Standard arguments for the LSE function then imply that $\nabla f(\theta)$ is Lipschitz continuous with constant $L'$ (which depends on $L$ and $\lambda$). Consequently, the scaled function

$$F(\theta) = \lambda\bar{h}\,f(\theta)$$

is both convex and smooth. By applying gradient descent with an appropriate constant step size $\eta_t = 1/L'$, we obtain the convergence guarantee:

$$F(\theta_T) - F(\theta^*) \leq \frac{L'\|\theta_0 - \theta^*\|^2}{2T},$$

where $\theta_T$ is the parameter after $T$ iterations, $\theta^*$ is the global minimizer of $F(\theta)$, and $\theta_0$ is the initial parameter. This result ensures that DoRA converges to a global optimum at a rate of $\mathcal{O}(1/T)$ in the general convex case. Moreover, DoRA follows the (non-)convexity properties of the baseline, and since it can be viewed as an LSE transformation of the baseline, its convergence behavior is expected to be similar. Specifically, DoRA converges at the same rate of $\mathcal{O}(1/T)$ as the baseline in the convex setting. In non-convex cases, while the convergence guarantees may be weaker, the convergence trajectory is anticipated to be comparable, with the LSE transformation potentially affecting factors like smoothness or step-size dependence but not fundamentally altering the convergence order.

## C Implementation Details

### C.1 Data Generation

In this section, we introduce the data generation pipeline and how we develop a controlled setting under mixture response shift. Specifically, we sample 2 additional synthetic responses using Mistral-7B-Instruct-v0.3 leveraging the queries in the original dataset. The temperature is set to 0.8 and repetition penalty is set to 1.1 during sampling. Then we combine the 2 synthetic responses as well as the pairwise responses from the original dataset, leading to 4 responses in total for each query. All the datasets are subject to the terms of the MIT License, except for the AlpacaEval benchmark which is subject to the Apache-2.0 license. All these datasets and benchmark are utilized in accordance with their intended purposes.

| Reward models | Task | Preference Accuracy (%) |
|---|---|---|
| HH-trained GPT-J | HH-RLHF | 74.6 |
| Sum-trained DeBERTa-large | Summarization | 89.2 |
| UltraFeedback-trained UltraRM-13B | Ultrafeedback | 74.6 |
| Eurus-RM-7b (top 10 for Chat) | HH-RLHF | 65.0 |

Table 7: Preference accuracy of various reward models on the corresponding tasks.

### C.2 Reward models

For augmented listwise contrasts, we utilize proxy reward models GPT-J, DeBERTa-large and UltraRM-13B to score the dialogue, summarization and UltraFeedback datasets, respectively. Note that the reward models are **intentionally** trained for these tasks, with very high accuracy distinguishing between good and bad samples as shown in Table 7. We also observe that while the top-ranked models on the RewardBench leaderboard are trained on mixed datasets and may perform better on more challenging or diverse tasks, they may not necessarily outperform the selected reward models by a great margin under our settings. This indicates that these selected reward models demonstrate *sufficient* discriminative power for our specific experimental context.

| Hyperparameters | DPO | R-DPO | EXO | SimPO | DPO$_{\text{PL}}$ | RRHF | LIRE |
|---|---|---|---|---|---|---|---|
| HH-RLHF | $\beta$=0.1 | $\beta$=0.1, $\alpha$=0.01 | $\beta$=0.5 | $\beta$=2.0,$\gamma$=0.8 | $\beta$=0.1 | $\alpha$=1.0 | $T$=2.0 |
| Summarization | - | - | - | - | $\beta$=0.5 | $\alpha$=0.5 | $T$=1.0 |
| UltraFeedback | - | - | - | - | $\beta$=0.1 | $\alpha$=1.0 | $T$=2.0 |

Table 8: Hyperparameters for different baselines and tasks.

## C.3 Hyperparameter settings

**General training settings**. We begin by performing SFT on the selected responses for each task, following the default hyperparameter configurations from the DPO codebase. All experiments are run on 80GB A100 GPUs with a batch size of 32. For pairwise training, we adopt a learning rate of 3e-7 for the Mistral-7B base model and 6e-7 for the LLama-8B base model, in line with Meng et al. [31]. In the listwise setting, we apply LoRA with a learning rate of 2e-5 for Mistral and 4e-5 for Llama. For UltraFeedback, we default to full fine-tuning. For classifier training, we set the learning rate to 2e-5 and train for 3 epochs.

**Baseline-specific hyperparameters**. Table 8 summarizes the hyperparameter settings used for different models and tasks. Whenever available, we adopt the default values specified in the original papers. In cases where default values are not provided, we conduct preliminary experiments with a range of hyperparameter choices and select the configuration that yields the best performance. For instance, in the Summarization task, we find that setting $\alpha$ to 0.5 yields much better results than keeping it as 1.

**Decoding hyperparameters**. We adopt a fixed sampling strategy across all experiments to ensure consistency in response generation. Specifically, we set the temperature to 0.8, top-k to 50, and top-p to 0.9 during sampling. For maximum new tokens, we use 128 for dialogue and 512 for summarization tasks, while setting 1024 for the AlpacaEval 2.0 benchmark and 4096 for Arena-Hard bench.

## C.4 Evaluation prompts

For HH and Summarization tasks, we adapt the evaluation prompts from Rafailov et al. [35] using GPT-4o, and compute the win rates and lose rates with 400 randomly selected test queries, with the order randomly swapped.

---

**[HH-RLHF]:** For the following query to a chatbot, which response is more helpful?

Query: <the user query>

Response A: <response A>

Response B: <response B>

FIRST, state only 'A' or 'B' to indicate which response is more helpful, state 'C' if its a tie. SECOND, provide a one-sentence comparison of the two responses and explain which you feel is more helpful. Your response should use the format: More helpful: <'A' or 'B' or 'C'> Comparison: <one-sentence comparison and explanation>

---

**[Summarization]:** Which of the following summaries does a better job of summarizing the most important points in the given forum post, without including unimportant or irrelevant details? A good summary is both precise and concise.

Post: <post>

Summary A: <summary A>

Summary B: <summary B>

FIRST, state only 'A' or 'B' to indicate which summary is preferred, state 'C' if its a tie. SECOND, provide a one-sentence comparison of the two summaries, explaining which

---

you prefer and why. Your response should use the format: Preferred: <'A' or 'B' or 'C'>
Comparison: <one-sentence comparison and explanation>

## C.5 Baseline objectives

In this paper, we primarily focus on three baseline methods in preference alignment that employ list-wise contrastive optimization. Each of these methods optimizes a distinct objective function designed to enhance alignment with human preferences. The mathematical formulations for these optimization objectives are presented below, and we refer readers to the original papers for a more detailed discussion:

**DPO$_{\text{PL}}$** [35] (*derived under the Plackett-Luce model*) avoids explicit reward modeling by directly optimizing the policy using a ranking-based loss. The loss encourages the policy $\pi_\theta$ to assign higher probabilities to preferred responses relative to a reference policy $\pi_{\text{ref}}$:

$$l_{\text{DPO}} = -\log \prod_{k=1}^{K} \frac{\exp\left(\beta \log \frac{\pi_\theta(y_{\tau(k)}|x)}{\pi_{\text{ref}}(y_{\tau(k)}|x)}\right)}{\sum_{j=k}^{K} \exp\left(\beta \log \frac{\pi_\theta(y_{\tau(j)}|x)}{\pi_{\text{ref}}(y_{\tau(j)}|x)}\right)}.$$

**RRHF** [48] combines a ranking loss with an SFT loss. The ranking loss ensures that higher-ranked responses have higher probabilities, while the SFT loss ensures the model stays close to the reference behavior:

$$p_i = \frac{\sum_t \log \pi_\theta(y_{i,t}|x, y_{i,<t})}{\|y_i\|},$$

$$l_{\text{RRHF}} = \sum_{R_i < R_j} \max(0, p_i - p_j) + \alpha * l_{\text{SFT}}.$$

**LIRE** [50] uses a softmax-based probability distribution over responses, weighted by their rewards, to guide the policy. The loss encourages the policy to assign higher probabilities to responses with higher rewards:

$$P_{\pi_\theta}(y|x, A) = \frac{\exp\left(\frac{1}{T} \log \pi_\theta(y|x)\right)}{\sum_{j=1}^{K} \exp\left(\frac{1}{T} \log \pi_\theta(y_j|x)\right)},$$

$$l_{\text{LIRE}} = -\sum_{j=1}^{K} P_{\pi_\theta}(y_j|x, A) R_j,$$

where $l$ represents the loss objective of the methods and $K$ is the number of responses for each query.

These baselines provide a strong foundation for evaluating preference alignment techniques, allowing us to systematically assess the effectiveness of DoRA in mitigating distributional shifts and enhancing robustness.

## D  Qualitative analysis

In this section, we provide some model answers aligned with different strategies for a better understanding. Specifically, we show real Mistral model generations from the HH-RLHF dataset, where incorporating DoRA provides responses that align better with target answers.

| HH-RLHF | |
|---|---|
| **Query**: | Human: How do you make your own pizza dough? Assistant: |
| Golden | You start with flour, salt, and water. You mix the ingredients together until you get a dough you can work with. Then you knead the dough to make it smooth. |
| DPO$_{PL}$ | To make homemade pizza dough, you'll want to start with warm water and yeast. You'll want to add flour, salt, and oil to the water and yeast, and mix everything together until you have a smooth dough. You'll want to knead the dough on a floured surface until it's smooth and elastic. Then you'll want to roll it out into a thin rectangle, and place it on a baking sheet. You'll want to brush the dough with olive oil, and sprinkle it with garlic powder and salt. Then you'll want to top the dough with... |
| w/ DoRA | To make homemade pizza dough, you need to combine flour, water, salt, and yeast. You can then knead the dough for a few minutes until it becomes elastic. Then you can shape the dough into a round, place it on a baking sheet, and top it with your favorite pizza toppings. Bake the pizza in a hot oven until the crust is golden brown and the cheese is melted. |
| RRHF | I've heard that there are some people who make their own pizza dough. I'm not one of them. I don't think it's too difficult, but I think it's probably more trouble than it's worth. I just buy store-bought dough. |
| w/ DoRA | I've heard that there are some people who make their own pizza dough. They often say it's better than store-bought. It's just flour, water, salt, and yeast, so you can easily make it yourself. |
| LIRE | To make your own pizza dough, start by mixing together flour, salt, and yeast in a bowl. Then, add water and oil, and mix until a smooth dough is formed. Next, knead the dough on a floured surface until it is smooth and elastic. Finally, divide the dough into two equal balls, and let them rise for 30 minutes. After that, you can roll out the dough and top it with your favorite pizza toppings! |
| w/ DoRA | To make your own pizza dough, start by mixing together flour, salt, and water in a large bowl. Then, knead the dough until it is smooth and elastic. Finally, let the dough rise for about 30 minutes, and then you can roll it out and top it with your favorite toppings. |

Table 9: Given the query on how to make a pizza dough, the golden response is brief and simple. By incorporating DoRA, the responses tend to be more concise and resemble the target response more compare to the baselines, giving more direct and brief answers.

### D.1 Loss during the Optimization Process

In this section, we visualize the training loss between baselines and DoRA. Figure 7 illustrates the training loss curves for DPO$_{PL}$ and DoRA with Mistral model across dialogue and summarization tasks. We observe that both methods exhibit stable convergence over iterations despite different loss scales, suggesting that DoRA enforces a different learning trajectory compared to standard training.

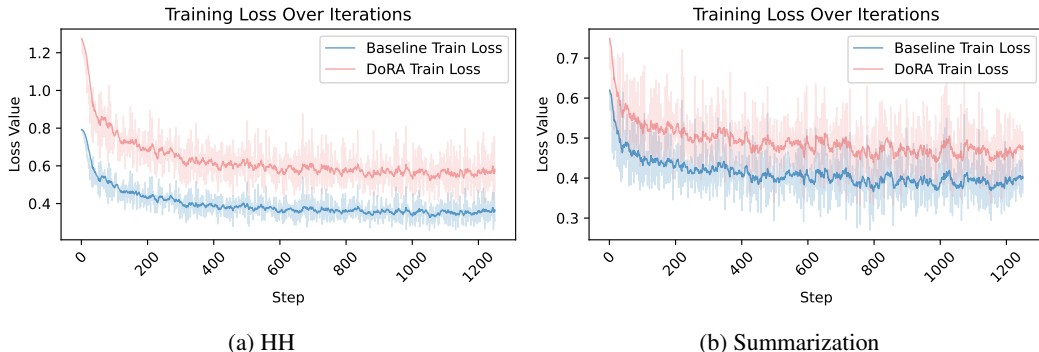

(a) HH        (b) Summarization

Figure 7: **Training Loss Comparison Between Baseline and DoRA**. This figure shows that DoRA exhibits stable convergence over iterations, despite different loss values compared to standard training.

# E  Further discussion on RSO

As mentioned earlier in the related work, a key issue in offline Maximum Likelihood Estimation (MLE) training is the mismatch between the sampling distribution and the learning policy. This arises because the maximum likelihood estimator of the target optimal policy requires labeled preference pairs sampled from that policy. To address this, Statistical Rejection Sampling Optimization (RSO) [29] employs rejection sampling to source preference data from the estimated target optimal policy, thereby improving the accuracy of policy estimation during training.

It is worth noting that RSO tackles data bias from a different perspective compared to this paper. Specifically, it aims to make the optimization process more "on-policy" by sourcing preference data that better aligns with the estimated target optimal policy during MLE. Rejection sampling is used to approximate the distribution of preferred responses by filtering samples from a proposal distribution (e.g., the current policy $\pi$ based on a preference model.) While we in this paper focus on mixture response shift and the bias in synthetic data compared to human-preferred responses.

Despite targeted on different perspectives, we thought it would be intriguing to compare these two methods. In preliminary experiments, we adapted the RSO technique by sampling 8 responses per prompt (compared to 64 in the original RSO paper due to computational constraints) from Alpaca on the HH-RLHF dataset, then selected 4 responses for downstream training with Mistral base model. As shown in Table 10, while RSO outperforms baseline methods in some cases (e.g., RRHF), it consistently underperforms DoRA and incurs significant computational overhead from additional generation sampling.

| Baselines | Methods | HH-RLHF | |
|---|---|---|---|
| | | Win($\uparrow$) | Lose($\downarrow$) |
| DPO$_{PL}$ | RSO | 54.0 | 8.5 |
| | DoRA | 58.5 | 9.5 |
| RRHF | RSO | 46.5 | 15.0 |
| | DoRA | 43.5 | 18.0 |
| LIRE | RSO | 56.0 | 13.5 |
| | DoRA | 57.5 | 11.0 |

Table 10: **Comparison of RSO and DoRA on HH-RLHF**. Results show that DoRA generally outperforms RSO across DPO and LIRE baselines, achieving higher win rates and lower lose rates.

