# OpenReview forum: "Leveraging robust optimization for llm alignment under distribution shifts"
_NeurIPS.cc/2025/Conference — NeurIPS 2025 poster_

### Official Review · Reviewer_8Cax · 2025-06-20

**Clarity:** 1
**Significance:** 2
**Originality:** 2
**Rating:** 2
**Confidence:** 3

**Summary:**

This paper proposes DoRA (Distributionally Robust Alignment), a new preference learning framework inspired by distributionally robust optimization (DRO), applied to LLM alignment. The key idea is to model preference optimization as minimizing regret over a worst-case distribution shift between the original preference dataset and the LLM’s evolving policy. To do this, the authors train classifiers to estimate the density ratio between the preferred and dispreferred responses, use those to compute importance weights, and apply a regularized regret loss to update the model. The method is evaluated primarily on the HH-RLHF dataset with LLaMA-7B, showing improved win rate over DPO and SimPO.

**Questions:**

Can the authors clearly define how the HH dataset win rate is computed? Is it from GPT-4 comparisons, pairwise preferences, or external rankers?

What specific role does the classifier play in estimating the density ratio? How is it trained and validated? More clarity and ablations are needed here.

Why is the method described as LLM-specific? Apart from the use of preference data from LLM outputs, the algorithm seems agnostic to language modeling.

Can the authors provide evaluation on more diverse benchmarks (e.g., MT-Bench, Arena-Hard, AlpacaEval)? One win rate metric is insufficient to support alignment claims.

Given the added complexity, how does DoRA perform under simpler ablations? For example, removing the density ratio weighting or using a simpler robustness penalty.

**Ethical Concerns:**

["NO or VERY MINOR ethics concerns only"]

**Limitations:**

yes

**Quality:**

1

**Strengths And Weaknesses:**

Strengths: The formulation is grounded in well-studied robust optimization theory.

Weaknesses:

Disconnect between method and LLM context:
Although the paper positions itself as a contribution to LLM alignment, the methodology appears to be a generic instance of distributionally robust optimization (DRO) applied to a preference dataset. The classifier-based density ratio estimation and reweighting procedure is standard in DRO literature, and the paper does not explain how the unique challenges of LLM alignment inform its algorithmic design. There is little adaptation to the sequential or structured nature of LLM outputs. As a result, the method feels like a transplant from a different domain rather than a carefully tailored alignment strategy.

Unclear evaluation metric:
The primary metric reported in the experiments is “HH dataset win rate,” but the paper fails to explain how this metric is computed. Is the win rate measured via GPT-4 preferences, a learned reward model, or some internal heuristic? What exactly constitutes a "win" in this setting? Against which model? Are responses sampled once per prompt or averaged over multiple runs? Without a clear description, it's hard to judge the reliability or fairness of this metric. Given that the win rate is the only reported outcome, this lack of transparency significantly weakens the empirical claims.

Limited empirical validation:
The evaluation is restricted to a single dataset (HH-RLHF) and a single win rate metric. No comparisons are made on other standard benchmarks used for LLM alignment evaluation, such as MT-Bench, AlpacaEval, Arena-Hard, or Open LLM Leaderboard metrics. Additionally, there is no breakdown of performance across prompt categories or qualitative error analysis. This narrow evaluation makes it hard to judge whether the proposed method offers any practical advantages in broader or more realistic alignment settings.

Unnecessarily complex and opaque setup:
The proposed method requires training separate classifiers to estimate the density ratio between preferred and dispreferred responses. These are then used to reweight training samples when optimizing the policy. However, the motivations for these design choices are not clearly justified, and the pipeline introduces substantial complexity. The use of SFT initialization is briefly mentioned but not analyzed. Overall, the system feels overengineered relative to the simplicity of existing approaches like DPO, and it is unclear whether the added complexity yields meaningful or robust benefits.

---

> ### Author Rebuttal · Authors · 2025-07-30
>
> Thanks for the questions, and we hope the following responses can address the confusion or misunderstanding.
>
> >$Q_1$: "Can the authors clearly define how the HH dataset win rate is computed? Is it from GPT-4 comparisons, pairwise preferences, or external rankers?"
>
> As noted in line 550, we use **GPT-4o** to compute win rates according to the provided evaluation prompt. To improve clarity, we will move this specification to the “Experimental Settings” section in the main paper.
>
> >$Q_2$: "What specific role does the classifier play in estimating the density ratio? How is it trained and validated? More clarity and ablations are needed here."
>
> The classifier provides scores used to compute the **calibration factor** in Equation (6). Training procedures, model architecture, and hyperparameters are detailed in **Algorithm 1** and **Appendix C.3 (line 536)**. Concretely, we train a BERT-base classifier using a standard binary classification loss, where the "chosen" response in each original dataset entry is labeled as 1 and all other responses as 0. We use a learning rate of 2e-5 and train for 3 epochs.
> We appreciate the suggestion for improved clarity and will make this training pipeline more explicit in the final version of the paper.
>
> > $Q_3$: "Why is the method described as LLM-specific? Apart from the use of preference data from LLM outputs, the algorithm seems agnostic to language modeling."
>
> Please note that we **do** **not** describe this method as LLM-specific. The framework is *method-agnostic* by design and is motivated as such throughout the paper. We also test it across different model architectures to demonstrate its generalization ability. We are happy to make further clarifications if needed.
>
> >$Q_4$: "Can the authors provide evaluation on more diverse benchmarks (e.g., MT-Bench, Arena-Hard, AlpacaEval)? One win rate metric is insufficient to support alignment claims."
>
> Thanks for asking. Please see **Table 3** in the main paper, where we provide results on Arena-Hard and AlpacaEval. The results suggest that DoRA consistently improves performance on diverse benchmarks.
>
> >$Q_5$: "Given the added complexity, how does DoRA perform under simpler ablations? For example, removing the density ratio weighting or using a simpler robustness penalty."
>
> Please note that the **calibration factor** is one of the core contributions to the proposed method; removing this, it falls back to a standard DRO baseline. We present such ablations in **Table 4 (Line 259)**, comparing DoRA against simpler, robustness-controlled variants such as vanilla DRO and reweighting-based optimization. These results show that DoRA achieves higher performance while maintaining stability.
>
> Thanks again for the questions, and we hope the above responses can mitigate the concerns raised. We are happy to make further clarifications and we welcome further discussion.

---

> > ### Comment · Area_Chair_zmzD · 2025-08-04
> > **Please respond to the author's rebuttal post**
> >
> > Hi Reviewer 8Cax, I see no response letting me know whether or not the rebuttal has changed
> > your opinion. You are the most negative review of this work and all the others have significantly higher scores. If you do not respond, I will have to discount your review. Could you please let me and the authors know by engaging? This process is critical to enabling the (S)ACs to make a decision on this work.
> >
> > --Your AC

---

> > ### Comment · Reviewer_8Cax · 2025-08-06
> >
> > Thank you for the detailed rebuttal and clarifications.
> >
> > I appreciate the explanation of the win rate computation and classifier training. Moving this information into the main text would improve clarity. The role of the classifier in computing the calibration factor is now clearer, and making the training pipeline more explicit would help readers better understand the method.
> >
> > Regarding $Q_3$, the authors note that the method is not LLM-specific and is intended to be general. If so, it is unclear why the evaluation is limited only to LLM preference data. Testing on other tasks or domains would help support the claim of generality.
> >
> > On the empirical side, while Table 3 includes more benchmark results, the evaluation remains narrow. The reported gains over baselines are minimal. For example, gaps as small as 0.1%, which fall well within the standard deviation (around 3%) on small datasets like AlpacaEval2 and Arena-Hard, which makes it hard to claim there is any improvement.
> >
> >
> > In summary, while the rebuttal clarifies several technical points, it does not fully resolve concerns about the novelty, evaluation scope, and practical relevance of the method. I will maintain my original rating.

---

> > > ### Author Response · Authors · 2025-08-06
> > >
> > > Thanks for your reply. First, we want to clarify that this paper focuses on "LLM Alignment" as the title suggests. The preference datasets (human/AI feedback) employed for both training and evaluation are a well-established and widely scrutinized benchmark family in this research area, ensuring comparability with prior work [1-5].
> > >
> > > Second, we would like to emphasize that this work focuses on robust optimization, as highlighted in the title. Our ultimate goal is to enhance the robustness of the alignment algorithm in terms of distributional shifts. To better illustrate how DoRA improves robustness under such cases, we use *self-training* and *label noise* as two representative stress tests (see the table below): 1. **Self-training** [6-8] – online updates lead to accumulated deviations between generated responses and the target distribution; 2. **Label noise** [1-2] – controlled injection of inaccuracies into preference labels (specified as a percentage).
> > >
> > > In both cases, DoRA consistently improves baseline performance to a great margin. We expect its value to be even greater in real-world or evolving deployment scenarios, where data often includes noisy synthetic outputs or imperfect annotations.
> > >
> > > || Self-training|       |     | Label noise             |              | |||
> > > | ------------ | ------------- |------------- |  ------------ | ------------------------------ | ------------ |------------ |------------ |------------ |
> > > | Dataset|HH-RLHF| | | AlpacaEval 2 |  |HH-RLHF | | |
> > > | Iter/Noise ratio |Iter 1|Iter 2|Iter 3 | 40% | 40%|20%|40%|60%|
> > > | Metric| Win rate| Win rate| Win rate| LC Win rate | Win rate |Win rate |Win rate |Win rate |
> > > | DPO          | 77.0|83.5| 85.0                 | 11.00                          | 10.99        |71.0 |64.5 |57.3 |
> > > | +DoRA        | 78.5|85.8|88.0 | 12.07                          | 12.06        | 74.5| 67.0| 60.8|
> > > | RRHF         | 45.8|47.8|50.5                      | 7.67                           | 4.84         |63.5 |42.3 | 26.3|
> > > | +DoRA        | 47.5|49.3|52.5                | 8.50                           | 5.51         | 65.5| 44.5| 30.8|
> > > | LIRE         | 80.3|83.0|84.5          | 26.52                          | 25.06        | 67.7|52.5 |61.4 |
> > > | +DoRA        |82.5|85.0|86.8                 | 27.65                          | 26.18        | 71.5| 55.0| 65.3|
> > >
> > > We will include these results in the final version. Thanks again for the feedback. We are glad to hear from you and welcome further discussion.
> > >
> > >
> > > [1]. Wu, Junkang, et al. "Towards robust alignment of language models: Distributionally robustifying direct preference optimization." arXiv preprint arXiv:2407.07880 (2024).
> > >
> > > [2]. Chowdhury, Sayak Ray, Anush Kini, and Nagarajan Natarajan. "Provably robust dpo: Aligning language models with noisy feedback." arXiv preprint arXiv:2403.00409 (2024).
> > >
> > > [3]. Rafailov, Rafael, et al. "Direct preference optimization: Your language model is secretly a reward model." Advances in neural information processing systems 36 (2023): 53728-53741.
> > >
> > > [4]. Meng, Yu, Mengzhou Xia, and Danqi Chen. "Simpo: Simple preference optimization with a reference-free reward." Advances in Neural Information Processing Systems 37 (2024): 124198-124235.
> > >
> > > [5]. Han, Jiaqi, et al. "$ f $-PO: Generalizing Preference Optimization with $ f $-divergence Minimization." arXiv preprint arXiv:2410.21662 (2024).
> > >
> > > [6]. Gulcehre, Caglar, et al. "Reinforced self-training (rest) for language modeling." arXiv preprint arXiv:2308.08998 (2023).
> > >
> > > [7]. Chen, Zixiang, et al. "Self-play fine-tuning converts weak language models to strong language models." arXiv preprint arXiv:2401.01335 (2024).
> > >
> > > [8]. Wu, Yue, et al. "Self-play preference optimization for language model alignment." arXiv preprint arXiv:2405.00675 (2024).

---

### Official Review · Reviewer_Jipr · 2025-06-26

**Clarity:** 4
**Significance:** 3
**Originality:** 3
**Rating:** 5
**Confidence:** 4

**Summary:**

This paper proposes DoRA (Distribution-aware optimization for Robust Alignment) to address distribution mismatch in preference learning. The core issue is that maximum likelihood estimators for optimal policies require samples from the target distribution, but training data typically contains a mixture of human-generated and LLM-generated responses from various sources.

DoRA uses distributionally robust optimization to handle this distribution shift. The method trains classifiers to identify the source distribution of each sample, then incorporates these predictions as importance weights in a robust optimization objective. This transforms the standard sum-of-log-exponentials loss into a log-sum-exponential form that minimizes worst-case expected loss. The approach is method-agnostic, working with various preference learning algorithms beyond DPO.

Experiments on HH-RLHF and UltraFeedback datasets using Mistral-7B and Llama-8B show improvements over baselines like Robust DPO and Dr.DPO across most settings.

**Questions:**

* (traditionally) Log-sum-exponential objectives are notoriously difficult to optimize, prone to numerical instability and vanishing gradients. While the authors add a 1/n stabilizing term, insufficient discussion of **practical** optimization considerations may hinder reproducibility
* Consider renaming the method, as "DoRA" is already used for a LoRA variant in parameter-efficient training.

**Ethical Concerns:**

["NO or VERY MINOR ethics concerns only"]

**Limitations:**

yes

**Quality:**

3

**Strengths And Weaknesses:**

## Strengths
- **Technically sound approach** with clear motivation addressing a practical problem in preference learning
- **Method-agnostic approach** that extends beyond DPO to various preference learning algorithms
- **Well-organized presentation** with focused contributions and thorough experimental evaluation

## Weakness
* **Limited practical gains vs. complexity**: Requires training separate BERT classifiers for each source distribution, adding significant implementation complexity for small improvements on ArenaHard and AlpacaEval 2.

---

> ### Author Rebuttal · Authors · 2025-07-30
>
> Thank you for the insightful comments and helpful suggestions!  We are grateful for the positive rating and hope that our responses alleviate the questions that were raised.
>
> > $Q_1$: Limited practical gains vs. complexity: Requires training separate BERT classifiers for each source distribution, adding significant implementation complexity for small improvements on ArenaHard and AlpacaEval 2.
>
> Thank you for raising this point. We acknowledge that DoRA introduces some additional implementation overhead due to training source-specific classifiers. However, these classifiers are lightweight, fast to train, and only trained once per source. Their outputs can be **pre-computed and cached**, resulting in **negligible overhead during policy optimization**.
>
> It’s also important to note that our current experiments are based on relatively clean datasets, which may understate DoRA’s practical utility. As a robustness control method, DoRA’s benefits become more apparent under *noisy or significantly-shifted distributions*. We show this in our stress tests (see the table below), including: 1. larger shifts in response (we use a model distributionally far from the target (Alpaca‑7B) to generate the synthetic responses. 2. larger label noise (preference labels are randomly corrupted in 40% of the training set).
>
> In both cases, DoRA consistently improves baseline performance on AlpacaEval 2. We expect its value to be even greater in real-world or evolving deployment scenarios, where data often includes noisy synthetic outputs or imperfect human annotations. Overall, DoRA provides strong robustness gains at a modest and manageable implementation cost.
>
> | AlpacaEval 2 | Larger shifts in response space |              | Larger label noise             |              |
> | ------------ | ------------------------------- | ------------ | ------------------------------ | ------------ |
> | **Methods**  | Length-controlled Win rate (%)  | Win rate (%) | Length-controlled Win rate (%) | Win rate (%) |
> | DPO          | 5.92                            | 5.50         | 11.00                          | 10.99        |
> | +DoRA        | 6.61                            | 6.03         | 12.07                          | 12.06        |
> | RRHF         | 3.16                            | 2.79         | 7.67                           | 4.84         |
> | +DoRA        | 4.73                            | 3.99         | 8.50                           | 5.51         |
> | LIRE         | 6.51                            | 7.47         | 26.52                          | 25.06        |
> | +DoRA        | 14.94                           | 11.75        | 27.65                          | 26.18        |
>
>
> We are happy to include this in the discussion and provide more ablation/analysis in the updated version. Thanks again for the feedback.
>
> > $Q_2$: "insufficient discussion of practical optimization considerations"
>
> We appreciate this thoughtful comment. To reduce the numerical instability of log-sum-exp–based objectives, we adopt a stabilizing term $\frac{1}{n}$ as an offset to the denominator (Line 149). This ensures that $\tilde{h}(z)$ is bounded in $(0,n)$, ensuring no single sub-distribution can dominate training. Besides, we also leverage the LSE trick by subtracting the max value before exponentiating to maintain numerical stability in practice. These techniques help stabilize training across all tasks in our experiments. We will include more details in the updated version, and we are happy to release the code to ensure reproducibility.
>
> >$Q_3$: "Consider renaming the method, as "DoRA" is already used for a LoRA variant in parameter-efficient training."
>
> Thanks for the kind advice. We will consider another name to avoid confusion in the updated version.
>
>  Thanks again for the questions, and we hope the above responses can mitigate the concerns raised. We are happy to make further clarifications and we welcome further discussion.

---

> > ### Comment · Area_Chair_zmzD · 2025-08-04
> > **Please respond to the author's rebuttal post**
> >
> > Hi Reviewer Jipr, I see no response letting me know whether or not the rebuttal has changed
> > your opinion. Could you please let me and the authors know by engaging? This process is critical to enabling the (S)ACs to make a decision on this work.
> >
> > --Your AC

---

> > ### Comment · Reviewer_Jipr · 2025-08-07
> >
> > Thank you for the detailed rebuttal and additional experiments. I appreciate your thorough responses to my concerns.
> >
> > The stress test results provide helpful context, but the improvements remain relatively modest. I maintain my score, as the paper makes a solid technical contribution with consistent, if modest, empirical improvements.

---

> > > ### Author Response · Authors · 2025-08-07
> > >
> > > Thank you for your review and for acknowledging our rebuttal efforts. We appreciate your recognition of the paper’s technical contribution and are grateful for your time and feedback. Best regards!

---

### Official Review · Reviewer_1ReX · 2025-06-29

**Clarity:** 3
**Significance:** 2
**Originality:** 2
**Rating:** 3
**Confidence:** 3

**Summary:**

The paper introduces DoRA (Distribution-aware Optimization for Robust Alignment), a framework aiming to address the LLM alignment in the presence of distribution shifts. Taking the modern trend of using synthetic preference data as an example, the paper propose the method that model remains resilient to distribution shifts between the training data and the target distribution while still benefiting from their scalability.
The authors propose a two-phase approach that first uses distribution classifiers to estimate the likelihood that training samples originate from the target distribution Q₀, then incorporates these calibration scores into a KL-divergence-based Distributionally Robust Optimization objective to reweight the loss function toward samples better aligned with the target distribution.
Empirical evaluation across multiple alignment tasks demonstrates that DoRA consistently improves model performance and win rates compared to various baseline approache

**Questions:**

- Could the authors elaborate more on the conceptual (and technical) novelty of introducing a sample-level classifier within the DRO framework? In particular, how does this differ fundamentally from prior work, and what capabilities does it enable beyond existing methods?
- How does DoRA perform under adversarial distribution shifts or highly corrupted synthetic data? An ablation study with listedwise dataset from LLMs of varying quality might be helpful.
- Although the author mentioned about the flexibility of the proposed method with diverse distributional shift, the author only demonstrated on listwise preference setting with synthetic response. Other source such as combining different datasource with different quality, online updates would help strengthen the contribution.
- Since DoRA emphasizes Q₀-labeled samples, how do the authors ensure that this does not amplify label noise or miscalibration? What happens when classifier calibration itself is unreliable?

**Ethical Concerns:**

["NO or VERY MINOR ethics concerns only"]

**Final Justification:**

The concern about noise from the newly introduced learned classifier is partially addressed by experimental results. The author demonstrates that the classifier's performance gains outweigh its drawbacks, supported by the rationale of the stabilizing factor and empirical results on Q2, Q4, and Q5. Assuming the proposed revisions and additional experiments are included in the final version, I am updating my score slightly more positively.

**Limitations:**

yes

**Paper Formatting Concerns:**

I didn't notice major formatting issues

**Quality:**

3

**Strengths And Weaknesses:**

### Strengths

- The proposed framework is well-motivated, addressing a crucial challenge in alignment training in the divergence between synthetic and target distributions.
- The expansion beyond traditional pairwise noise robustness to broader distributional concerns is a timely and necessary shift in the alignment literature.
- The proposed method is method-agnostic and can be integrated into diverse alignment objectives, including different synthetic sources or online updates
- The writing is generally clear and well-structured. Key concepts such as mixture response shift, KL-DRO, and the calibration mechanism are explained clearly. The proposed method is demonstrated in a both principled and practical way.
- Experiments span pairwise and listwise settings across multiple datasets and models, showing consistent (although sometimes modest) gains in alignment quality.

### Weaknesses
- The contribution of introducing new sample-level classifier on existing method DRO offers limited novelty and add marginal contributions.
- While the paper provides theoretical motivation, the empirical gains are sometimes marginal (e.g., +0.6% in win rate for DPO with Mistral-7B), raising questions about cost-effectiveness. The added complexity of training additional classifiers and additional inference might not be justified.
- There is insufficient clarity on how DoRA’s calibration is different from reward models. Both attempt to score sample quality and may collapse to a combined reward models. The classifier based approach still hold the limitation of reward models that are prone to biases, potentially leading to suboptimal labelling. The authors should discuss this distinction more clearly.
- The approach might amplify label noise rather than mitigate it, since it emphasizes Q₀ without validating whether those labels are truly correct.

Minor feedback on Clarity:
- The details about win rate (e.g. using GPT-4o) is introduced first in Table 2. Earlier context would help orient readers.
- Figure 1 misleadingly depicts $Q_₁$ as easily separable from $Q_₀$, contradicting evidence that modern LLMs approximate human preferences well. In addition, unclear definition of $Q_1$ makes P unclear in Figure1, which prevent it from delivering the clear picture of overall method and its strength.
- notation of $h(z)$ vs. $h^{\sim} (z)$ is confusing, as one refers to the Radon-Nikodym derivative and the other to the calibration term.

---

> ### Author Rebuttal · Authors · 2025-07-30
>
> Thanks for the insightful comments and helpful suggestions! We hope that our responses help alleviate the concerns that were raised.
>
> > $Q_1$: "The contribution of new sample-level classifier on existing DRO framework"
>
> Thanks for asking. Our method is not merely a straightforward extension of DRO by adding a classifier. Instead, it introduces a fundamentally new formulation and training objective for robust alignment under synthetic data mixtures:
>
> (i) **Problem reframing**: We step beyond traditional empirical risk minimization (ERM) approaches used in alignment, and formally define a new robustness setting: mixture response shift. This setting, though increasingly common in practice due to the prevalence of synthetic data, has been *under-explored* in the literature.
>
> (ii) **Robust objectives with calibration factor**: Building on this, we derive the robust objective for existing alignment methods by introducing a *calibration factor* into the uncertainty set, where the probabilistic classifiers are leveraged to compute these factors.
>
> (iii) **Closed-form solution with over-pessimism mitigation:** The resulting objective admits a *closed-form dual solution*, and directly mitigates the over-pessimism commonly observed in classical DRO, where the model can over-focus on outliers or noisy examples.
>
> We also conclude some key differences with previous DRO literature in the following table.
>
> | Method              | Robust to                                                    | Optimization Focus                                           | Granularity of trust | pairwise & listwise? | Over‑pessimism control                        |
> | ------------------- | ------------------------------------------------------------ | ------------------------------------------------------------ | -------------------- | -------------------- | --------------------------------------------- |
> | **Standard KL‑DRO** | General distribution shift          | Worst-case loss over the entire training distribution        | Global               | Pairwise only        | No (often over‑pessimistic)                   |
> | **CVaR-DRO**        | Subpopulation shift in covariates ($\mathcal{X}$)            | Worst-case loss over any $\alpha-$sized subset of the training set | Per‑group            | Pairwise only        | Limited (moderates extremes but still coarse) |
> | **DoRA (ours)**     | Response/output shift in $\mathcal{Y}$ (mixture of output distributions per query) | Worst-case loss over target distribution alignment           | Per‑sample           | Yes                  | Yes                                           |
>
> > $Q_2$: "Performance under adversarial distribution shifts/highly corrupted synthetic data?"
>
> This is a good point. We focus on “clean” benchmark datasets (no synthetic corruptions) in the main paper, but we are happy to add the following corrupted-label experiments to serve as stress tests to confirm DoRA’s ability under adversarial shifts. Specifically, we introduce corruption to the data by randomly flipping the labels. Experiments show that incorporating DoRA consistently improves robustness under increasing label corruption, yielding consistent gains on different benchmarks.
>
>    | Dataset               | HH       |          |          | AlpacaEval 2 |          |
>    | --------------------- | -------- | -------- | -------- | ------------ | -------- |
>    | Corruption rate| 20%      | 40%      | 60%      | 40%          | 40%      |
>    | Metrics| Win rate (%)| Win rate (%)| Win rate (%)| Length-controlled Win rate (%) | Win rate (%)|
>    | DPO | 71.0     | 64.5     | 57.3     | 11.0         | 11.0     |
>    | +DoRA| 74.5     | 67.0     | 60.8     | 12.1         | 12.1     |
>    | RRHF | 63.5     | 42.3     | 26.3     | 7.7          | 4.8      |
>    | +DoRA  |65.5     | 44.5     | 30.8     | 8.5          | 5.5      |
>    | LIRE    | 67.7     | 52.5     | 61.4     | 26.5         | 25.1     |
>    | +DoRA   | 71.5     | 55.0     | 65.3     | 27.7         | 26.2     |
>
>  We will add these new experimental results to the revised manuscript and highlight them in the discussion.
>
> >$Q_3$: "how DoRA’s calibration is different from reward models."
>
> DoRA's calibration is fundamentally different from reward models, and the two operate on **different axes** of the alignment pipeline. First, DoRA relies on a learned probabilistic binary classifier which estimates *distribution membership*, not *human-preference utility*. While reward models directly score “how good a sample is”, DoRA’s classifier scores "how likely a sample comes from the target distribution". Therefore, DoRA’s calibration factor is a **density ratio proxy**, not a reward score. Its role is to modulate the DRO uncertainty set (to determine how much robustness weight to assign to each sample).
>
>  We agree that any learned model can be biased if trained on unrepresentative or mislabeled data. However, reward model bias directly impacts what the model learns to generate. If it reflects skewed or noisy preferences, it can misdirect optimization and lead to alignment failures.
> In contrast, our classifier only **distinguishes between sources** (e.g., human vs. synthetic), enabling selective robustness without steering generation itself. Moreover, the impact of classifier miscalibration is **bounded**.
>    As we describe in Line 146 and Eq. (6), we introduce a *stabilizing factor* (e.g., a small additive term in the denominator) that prevents any individual sample from dominating the optimization. This ensures that even overconfident classifier outputs do not result in extremely unstable or biased learning.
>
>  >$Q_4$:"performance on combining different datasource with different quality, online updates"
>
> Thanks for raising this point. While the primary **focus** of this paper is on **mixture response shift with synthetic data** in offline training, we briefly mention *online updates* as a promising direction in “Limitation and Future Work” section.
>
>  To illustrate DoRA’s robustness in such scenarios, we conduct a small-scale experiment: we train LLama3.2‑3B on the HH dataset (Iter 1), then continue training on its own generations for two more iterations. This setup induces an evolving mixture shift as the response distribution becomes increasingly synthetic. The results show that DoRA remains effective under this shift, suggesting that its instance-level calibration also generalizes to online or self-generated data settings, where distribution drift occurs naturally.
>
>    |       | Iter 1 | Iter 2 | Iter 3 |
>    | ----- | ------ | ------ | ------ |
>    | DPO   | 77.0   | 83.5   | 85.0   |
>    | +DoRA | 78.5   | 85.8   | 88.0   |
>    | RRHF  | 45.8   | 47.8   | 50.5   |
>    | +DoRA | 47.5   | 49.3   | 52.5   |
>    | LIRE  | 80.3   | 83.0   | 84.5   |
>    | +DoRA | 82.5   | 85.0   | 86.8   |
>
> >$Q_5$: "how do the authors ensure that this does not amplify label noise or miscalibration? What happens when classifier calibration itself is unreliable?"
>
> Thanks for asking, and we try to answer this from the following two perspectives.
>
> 1. "Label noise": We first clarify that $Q_0$ denotes the **target distribution** (trusted human-preferred responses from **known** sources/ground truth), while label noise in the alignment context typically refers to incorrect “chosen/rejected” annotations.
> From this perspective, if label noise exists in the *original dataset*, DoRA can actually **mitigate its impact** by downweighting those samples that are distributionally misaligned with the target (i.e., with low calibration scores).
>
> 2. "Classifier accuracy": If the concern instead refers to errors in the classifier itself, we agree that any learned model may be imperfect. However, we observe that our lightweight classifiers achieve consistently high accuracy across datasets, as shown below, indicating reliable calibration in practice:
>
> |  Dataset        | HH-RLHF | Summarization | UltraFeedback |
> | --------- | ------- | ------------- | ------------- |
> | Test Acc. | 94.8%   | 93.9%         | 92.7%         |
>
>
>
> >"Minor feedback on Clarity"
>
>    1. Thanks for pointing this out and we will place the explanation of "Win Rates" earlier in the main paper.
>
>    2. >"misleadingly depicts  $Q_1$ as easily separable from $Q_0$, contradicting evidence that modern LLMs approximate human preferences well."
>
> Thank you for the feedback. We clarify that **Figure 1 illustrates distributional separability, not preference similarity.** While it is true that modern LLMs (e.g., GPT‑4) may approximate human preferences in aggregate utility—as reflected by similar reward scores—this does **not** imply that their response distributions are identical.
> In fact, we find that $Q_1$ (synthetic responses) and $Q_0$ (human-preferred responses) often exhibit distinct generation patterns, which are learnable by a classifier with over 90% accuracy in our setting. This supports the distribution-level separability shown in Figure 1.
>
> We will revise Figure 1 to (i) explicitly clarify the meaning of $Q_1$ and $P$ in the caption and (ii) adjust the visual to better reflect realistic overlaps while still communicating the method’s key intuition.
>
>   3. >"notation of $h(z)$ vs. $\tilde{h}(z)$ is confusing..."
>
> We clarify that $h(z)$ denotes the Radon-Nikodym derivative $\frac{d Q_0}{dQ}$, which expresses the ideal density ratio between the two distributions, but is generally *incomputable in practice*. The symbol $\tilde{h}(z)$ refers to our *practical, learned approximation* of $h(z)$, obtained via a calibrated probabilistic classifier.
>
>   Thus, the notation distinction emphasizes that $\tilde{h}(z)$ is an implementable surrogate, not the true derivative. We will clarify this in the final version by explicitly linking the two in notation and explanation.
>
> Thanks again for the questions, and we hope the above responses can address the confusion or misunderstanding. We are happy to make further clarifications and we welcome further discussion.

---

> > ### Comment · Area_Chair_zmzD · 2025-08-04
> > **Please respond to the author's rebuttal post**
> >
> > Hi Reviewer 1ReX, I see no response letting me know whether or not the rebuttal has changed your opinion. Could you please let me and the authors know by engaging? This process is critical to enabling the (S)ACs to make a decision on this work.
> >
> > --Your AC

---

### Official Review · Reviewer_2E9q · 2025-06-30

**Clarity:** 3
**Significance:** 3
**Originality:** 3
**Rating:** 4
**Confidence:** 4

**Summary:**

The paper consider the problem of robust preference alignment in LLM. The approach is to first learn a classifier to assign a calibration value to each training sample, and then leverage it to the target human-preferred distribution. The objective is formulated as a distributionally robust optimization with respect to KL divergence.

**Questions:**

1. Construction of $\tilde h(z)$ is not completely clear to me. How does one compute the constants $\gamma_j$ and $\beta_j$?

2. How well does this method generalize to datasets not used for training? Did the authors test the aligned model on a dataset that has not been used for training (not considering train-test split)?

3. I see that golden/reference distribution is approximated using trusted human-labeled data. Clarifying questions, is the "golden" distribution same as the target/preferred distribution? How one may obtain such a data?

**Ethical Concerns:**

["NO or VERY MINOR ethics concerns only"]

**Final Justification:**

The authors have address my comments, so I have raised my evaluation score.

**Limitations:**

Please see the **Strengths And Weaknesses** and **Questions** sections.

**Quality:**

3

**Strengths And Weaknesses:**

**Strengths:** The paper consider a distribution-aware optimization framework to robustly align LLM outputs with human preferences under distribution shifts. The framework relies on the mixture response shift assumption, which posits that the training data comprises a mixture of heterogeneous sub-distributions. Then the procedure is broken into two parts: 1. employ a calibration score to each training sample, estimating its alignment to human preferences, and 2. integrate the calibration scores to a distribution-aware optimization objective that minimizes the worst case loss.

**Weaknesses:** I have some concerns about clarity of the expositions and generalizability of the experimental results. Please see my questions in the **Questions** section.

---

> ### Author Rebuttal · Authors · 2025-07-30
>
> Thank you for the insightful comments and helpful suggestions! We hope that our responses below help alleviate the concerns that were raised.
>
> > $Q_1$. "Construction of $\tilde{h}(z)$ is not completely clear to me. How does one compute the constants $\gamma_j$ and $\beta_j$ ?"
>
> Thanks for the detailed review. We clarify the definitions of $\beta_j$ and $\gamma_j$ as follows:
>
>   $\beta_j$ denotes the **relative propotion** of sub-distribution $Q_j$ in the dataset;
>
>  $\gamma_j$ is defined as the **imbalance ratio** between a synthetic distribution and the golden distribution $\frac{q(y=0)}{q({y=1})}=\frac{\beta_j}{\alpha}$.
>
>    For example, suppose for a given query, the responses set includes: 1 response from golden distribution $Q_0$, 2 responses from $Q_1$, and 3 responses from $Q_2$, then $\beta_1=\frac{2}{1+2+3}=\frac{1}{3},\gamma_1$ = $\frac{\beta_1}{\alpha}=\frac{\frac{1}{3}}{\frac{1}{1+2+3}}=2$. Then $\tilde{h}(z)$ can be computed according to Eq. (6), where $c_{\phi_j}$ represents the score from the learned classifier for sub-distribution $Q_j$.
>
> >$Q_2$: "How well does this method generalize to datasets not used for training? ..."
>
> Thank you for the insightful question. **Table 3 (line214)** reports results on **AlpacaEval 2** and **Arena-Hard** using models trained on **UltraFeedback**, where the test queries come from a different distribution to the training data. These results demonstrate that DoRA generalizes effectively when evaluated out-of-distribution.
>
> We *further* evaluate generalization capability under the following scenarios:
> (1) Training on HH-RLHF and testing on summarization (different tasks), and
> (2) Training with label noise (40% of labels randomly shuffled).
> As shown in the table below, DoRA improves baseline performance in all cases, demonstrating robust generalization.
>
> | Training data | HH-RLHF       | UltraFeedback (w/ noisy label) |              |
> | ------------- | ------------- | ------------------------------ | ------------ |
> | Test Data | Summarization | AlpacaEval 2                   |              |
> | Metrics       | Win rate(%)   | Length-controlled Win rate (%) | Win rate (%) |
> | DPO           | 36.3          | 11.00                          | 10.99        |
> | +DoRA         | 39.0          | 12.07                          | 12.06        |
> | RRHF          | 23.5          | 7.67                           | 4.84         |
> | +DoRA         | 27.5          | 8.50                           | 5.51         |
> | LIRE          | 43.8          | 26.52                          | 25.06        |
> | +DoRA         | 49.5          | 27.65                          | 26.18        |
>
> >$Q_3$. "Clarifying questions, is the "golden" distribution same as the target/preferred distribution? How one may obtain such a data?"
>
> Yes, in our setting, the “golden” distribution corresponds to the positive class when training the probabilistic classifiers. It is constructed with samples drawn from the target/preferred distribution $Q_0$.
> In practice, we approximate this golden distribution using "chosen" responses from trusted, high-quality preference datasets, where preferences are made either by humans or by a very strong LLM such as GPT‑4. These responses serve as a reasonable proxy for the target distribution.
>
>    Thank you again for the detailed review and insightful feedback, and we will include these clarifications in the revised manuscript. We believe your input has already helped improve the paper and look forward to engaging with you further during the discussion period.

---

> > ### Comment · Area_Chair_zmzD · 2025-08-04
> > **Please respond to the author's rebuttal post**
> >
> > Hi Reviewer 2E9q, I see no response letting me know whether or not the rebuttal has changed your opinion. Could you please let me and the authors know by engaging? This process is critical to enabling the (S)ACs to make a decision on this work.
> >
> > --Your AC

---

### Note · Authors · 2025-08-12

We thank the reviewers for their thoughtful evaluations and constructive feedback. Across the reviews, the overall consensus on the merits of our work includes:  (1) addresses a key challenge in robust preference alignment under distribution shifts, (2) is method-agnostic and broadly applicable beyond DPO, and (3) is supported by a well-motivated theoretical foundation and extensive empirical validation. Reviewers generally appreciated the clarity of presentation, the principled KL-based distributionally robust optimization formulation, and the rigorous experimental design spanning multiple datasets, models, and alignment objectives.

The primary concerns raised focused on: (1) DoRA’s generalization to unseen datasets, (2) the modest empirical gains in certain cases, (3) robustness under adversarial shifts or online updates, and (4) the calibration mechanism’s reliability. Through detailed clarifications and additional experiments, we demonstrated that DoRA achieves consistent and stronger improvements under label noise and in self-play scenarios. We are grateful that most reviewers found these responses satisfactory and acknowledged the solid technical contribution of our work. While we regret that Reviewer 8Cax did not engage further on question (2) regarding modest empirical gains, we appreciate the constructive dialogue from all reviewers and will incorporate their feedback into the updated manuscript.

Overall, we believe that this paper offers a valuable and principled contribution to the alignment literature, with a robust formulation and practical relevance in scenarios involving heterogeneous synthetic data. We deeply appreciate the reviewers’ constructive engagement, as their feedback has significantly enhanced the quality of this work.

---

### Decision · Program_Chairs · 2025-09-17

**Decision:**

Accept (poster)

**Comment:**

This work proposes a method to ensure more robust alignment for LLMs. The authors first seek to measure a (pseudo) gold standard target distribution and then construct classifiers that attempts to inform via a score whether or not an arbitrary preference sample maps reasonably onto that distribution. Summarizing the reviewer's discussions.

Pros:
- The idea is theoretically sound and is timely given that most mainstream alignment approaches focus on combining all human value data into a single size fits all system. Being target distribution aware at train time itself lends itself more easily to approaches such as personalization and fitting to pluralistic values across different pools of humans
- Practical benefits over established techniques like DRO seem significant
- DoRA as a method can compose with many classes of online and offline algorithms, seemingly with not too great changes required for existing infra. This is important as it means it is easier for the wider community to adopt.

Cons:
- The obvious fundamental concern is that this method is restricted to only those instances where you can reasonably measure the target distribution in the first place, in many cases this isn't possible - and I'd like to see an acknowledgement from the authors pointing out when and when to not attempt to use DoRA.
- Some of the empirical gains in specific settings (e.g. Arena Hard) are minimal, especially given somewhat outdated datasets/benchmarks used such as HH-RLHF and Alpaca Eval 2. Some of these concerns appear to have been addressed during rebuttals

Overall, the authors have made a good faith effort to address concerns during the rebuttal period and some reviewers like 2E9q were swayed to lean accept. After discounting a review that did not engage (8Cax), I am also inclined to lean towards an accept as the pros outweigh the cons for this work.